# EDITLENS: QUANTIFYING THE EXTENT OF AI EDITING IN TEXT

**Katherine Thai**[1,2]   **Bradley Emi**[1]   **Elyas Masrour**[1]   **Mohit Iyyer**[3]
[1]Pangram Labs   [2]University of Massachusetts Amherst   [3]University of Maryland, College Park

## ABSTRACT

A significant proportion of queries to large language models ask them to *edit* user-provided text, rather than generate new text from scratch. While previous work focuses on detecting fully AI-generated text, we demonstrate that AI-edited text is distinguishable from human-written and AI-generated text. First, we propose using lightweight similarity metrics to quantify the magnitude of AI editing present in a text given the original human-written text and validate these metrics with human annotators. Using these similarity metrics as intermediate supervision, we then train EDITLENS, a regression model that predicts the amount of AI editing present within a text. Our model achieves state-of-the-art performance on both binary (F1=94.7%) and ternary (F1=90.4%) classification tasks in distinguishing human, AI, and mixed writing. Not only do we show that AI-edited text can be detected, but also that the degree of change made by AI to human writing can be detected, which has implications for authorship attribution, education, and policy. Finally, as a case study, we use our model to analyze the effects of AI-edits applied by Grammarly, a popular writing assistance tool. To encourage further research, we commit to publicly releasing our dataset and models.

⬡ github.com/pangramlabs/EditLens

## 1 INTRODUCTION

Large language models (LLMs) generate text that is difficult to distinguish from human writing, enabling malicious applications such as academic plagiarism and fake review farms, thus motivating the need for accurate AI detection. While existing detectors frame the task as binary classification (fully human vs. fully AI-generated), mainstream LLM usage increasingly involves *co-writing*, where LLMs are used for editing and brainstorming via services like Grammarly,[1] Sudowrite,[2] or Google Docs' Gemini integration. In fact, a recent OpenAI study of over 1M ChatGPT conversations (Chatterji et al., 2025) shows that "about two-thirds of all Writing messages ask ChatGPT to modify user text (editing, critiquing, translating, etc.) rather than creating new text from scratch." Binary AI detection systems are not well-suited to detect such mixed-authorship texts: for example, Saha & Feizi (2025) find that binary detectors often flag AI-polished text as AI-generated, limiting their utility in situations where light AI editing is acceptable but fully AI-generated text is not.

In this paper, we develop EDITLENS, the first AI detector that estimates the extent of AI editing in a text as a continuous score. Previous work on detecting mixed AI and human text has treated the task as either a boundary detection problem (Kushnareva et al., 2024; Lei et al., 2025), a sentence-wise classification task (Wang et al., 2023), or a ternary classification problem between human, AI, and mixed text (Abassy et al., 2024; Wang et al., 2025). However, modern collaborative editing involves layered revisions, suggestions, and refinements that blur traditional notions of authorship, making it challenging to definitively attribute specific segments to either human or AI authors and rendering boundary detection and sentence-level tasks ill-posed. Although the ternary classification approach does not require assigning direct authorship to discrete segments, it is unable to quantify the degree or the magnitude of AI editing: Was the text lightly edited for spelling and grammar, or completely

---

[1]https://www.grammarly.com/
[2]https://sudowrite.com/

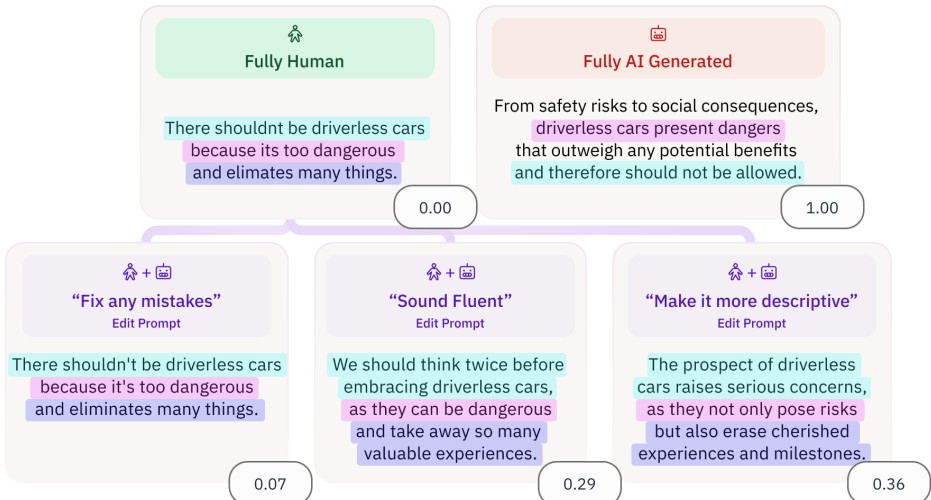

Figure 1: AI edits exist on a continuous spectrum from fully human written to fully AI generated. Here we show three versions of the same human-written text after different edits have been applied by an LLM alongside the cosine distance between the edited text and the fully human text. Texts have been truncated for space. "Fix any mistakes," the most mild edit according to cosine distance, results in a text with only spelling and grammar errors corrected, while "Make it more descriptive" closely adheres to the ideas in the human-written text while substantially rewriting it.

rewritten and restructured? Rather than classifying a text category, our model **directly regresses a score that indicates the degree of AI involvement in the production of the text as a whole.**

Our contributions are the following:

1. We introduce a comprehensive dataset spanning a full taxonomy of AI-edits to human-written texts.

2. We quantify the amount of AI editing applied to each text via lightweight similarity metrics, and validate that the similarity metrics correlate with the judgments of expert human annotators trained to detect AI writing styles.

3. We use these similarity metrics to finetune a regression head on an open-source large language model to detect the amount of AI-editing present given only the edited text.

4. When converted from a regression model to a binary or ternary classification model, we show that our model, EDITLENS, achieves state of the art performance, outperforming the best binary classifiers by 8%, and outperforming the best ternary classifiers by 16% (macro-F1).

5. We also show that unlike the discrete classifiers, the regression model is able to show nuance in progressively classifying more intense edits with higher scores, with case studies on APT-Eval, Beemo, and Grammarly.

Our findings have wide-ranging implications for AI text detection policy. By enabling measurement for the level of AI involvement, more flexible policies acceptable usage of generative AI models can be consistently enforced. Furthermore, our work can help mitigate false positives, a critical limitation of existing binary AI text classifiers. With the ability to control the amount of AI editing allowed, a much lower false positive rate can be achieved under the policy cap framework suggested by Jabarian & Imas (2025) for implementation in high-stakes settings such as academic integrity.

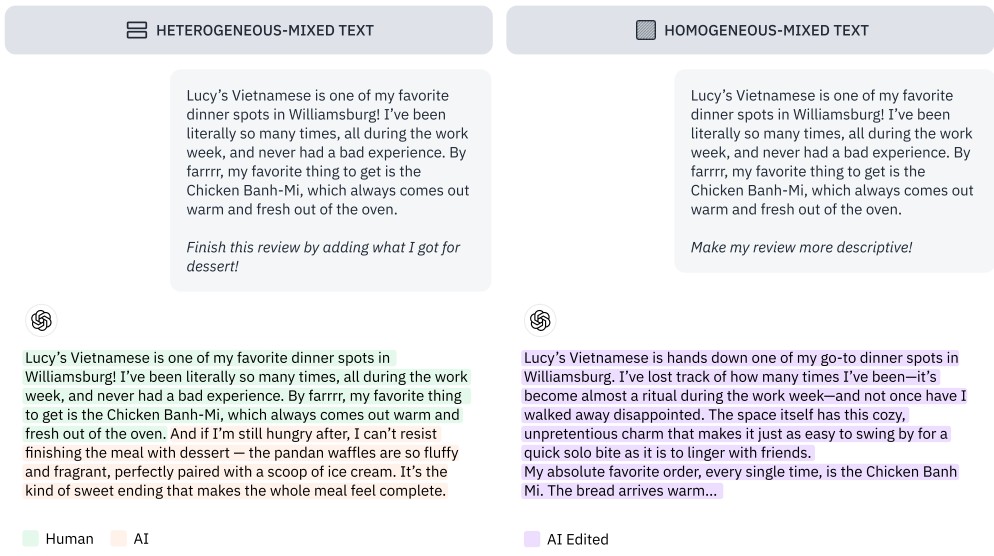

Figure 2: Examples of heterogeneous and homogeneous mixed authorship texts. In heterogeneous mixed text, authorship of each token is clearly attributable. But in homogeneous mixed text, the human-originated ideas are clearly present in each rewritten sentence by the model, making it impossible to assign binary labels of authorship to any word or sentence.

# 2 QUANTIFYING AI EDIT MAGNITUDE

## 2.1 HOMOGENEOUS VS. HETEROGENEOUS MIXED AUTHORSHIP

To better motivate our work, we first introduce the concepts of **heterogeneous** and **homogeneous** mixed authorship texts.

In the heterogeneous case, authorship of each segment of text can be directly attributed to a human or AI. An example of this is a situation where a human writes one paragraph and asks the AI to write the following paragraph. In cases like this, there exist one or more boundaries between human and AI segments. One can create token-level labels for heterogeneous mixed texts: every token was authored by either human or AI. Heterogeneous mixed text detection (also called fine-grained AI text detection) has been previously studied by Kushnareva et al. (2024), Wang et al. (2023), and Lei et al. (2025).

In the homogeneous case, authorship is entangled by the editing process. An example of this is a situation where a human writes a paragraph and asks an AI to paraphrase it. Even if AI replaces every word in the paragraph with a synonym, authorship is still mixed. As such, token-level binary labels are insufficient measures of authorship in this case, as both parties have provided input throughout the entire document. Despite its increasing prevalence, homogeneous mixed AI text is understudied, and we focus the rest of the paper on detecting this kind of mixed text.

## 2.2 TASK DEFINITION: HOMOGENEOUS MIXED TEXT

In many practical scenarios, a human-written document $x$ is subsequently edited to yield a new document $y$, where *multiple* sequential edits may have been performed by one or more agents (human or AI) in an indistinguishable fashion to produce $y$. Unlike the heterogeneous mixed-text setting, where each segment is assumed to be authored wholly by either a human or an LLM, here authorship is *latent and entangled within the editing process*. Our objective is not to attribute authorship, but to *predict the magnitude of change* between $x$ and $y$ according to a similarity metric that agrees with expert judgments of the magnitude of AI writing style and semantics.

We model the edited text as the image of an editing operator $\mathcal{E}_\lambda$ applied to $x$:

$$y \;=\; \mathcal{E}_\lambda(x;z), \qquad z \sim p(z), \quad \lambda \in \Lambda,$$

where $z$ denotes a (latent) sequence of micro-edits (insertions, deletions, substitutions, reorderings) possibly performed by a mixture of editor types (humans or AIs) and $\lambda$ summarizes an *edit intensity*. In the homogeneous setting, the editor identity within $z$ is unobserved and not required at training or inference time. For simplicity, in this study, we focus on the case where a human text is edited in one pass by a single AI language model, but we also present results for multiple passes, and human-edited AI text as case studies in generalization.

**Similarity-driven target.** Let $\mathrm{sim} : \mathcal{X} \times \mathcal{X} \to [0,1]$ be a fixed similarity functional. We define a change magnitude functional $\Delta : \mathcal{X} \times \mathcal{X} \to [0,1]$ by a monotone transformation of similarity (or distance):

$$\Delta(x,y) \;=\; g\big(\mathrm{sim}(x,y)\big), \quad \text{e.g.,} \quad g(s) = 1 - s$$

where $\mathrm{sim}$ is a nonnegative distance. $\Delta(x,y) = 0$ for identical texts (no edits) and increases as heavier editing is applied to form $y$. We motivate the particular choice of $\mathrm{sim}$ below via agreement with expert annotators' perception of the amount of AI pervasiveness within a text, and it is assumed known during training and evaluation.

**Inference with edited text only.** In most practical settings, only the edited document $y$ is available at inference time. We therefore learn a *single-input* predictor that maps $y$ directly to a change magnitude without reconstructing or retrieving a source $x$:

$$f_\theta^{\mathrm{ssi}} : \mathcal{X} \to [0,1], \qquad \hat{\Delta}(y) = f_\theta^{\mathrm{ssi}}(y).$$

Training remains *supervised* using pairs $\{(x^{(i)}, y^{(i)})\}_{i=1}^{N}$ only to compute targets $\Delta^{(i)} = \Delta(x^{(i)}, y^{(i)})$; the model never conditions on $x$ at inference. Concretely, we optimize

$$\min_\theta \; \frac{1}{N} \sum_{i=1}^{N} \mathcal{L}\Big( f_\theta^{\mathrm{ssi}}\big(y^{(i)}\big), \, \Delta\big(x^{(i)}, y^{(i)}\big) \Big).$$

The Bayes-optimal predictor for this objective is the conditional expectation

$$f^\star(y) \;=\; \mathbb{E}[\Delta(X,y) \mid Y = y],$$

but crucially we *do not* estimate this expectation via reconstruction of $x$. Instead, $f_\theta^{\mathrm{ssi}}$ learns discriminatively from $y$ alone, absorbing the necessary inductive biases (e.g., lexical volatility, style drift, fluency/consistency cues) to approximate $f^\star$ from labeled examples.

For additional discussion of the precise differences between homogeneous and heterogeneous mixed detection formulations, see the Appendix.

## 3 Training a model to detect AI edits

### 3.1 Creating a Homogeneous Mixed Text Dataset

Because no dataset of homogeneous mixed AI-generated text exists *at scale*, we create a training set for this task.

We begin by collecting a source dataset of fully-human and fully-AI-generated texts. We select human-written texts from prior to the release of large language models in 2022 from 4 domains: reviews from Amazon (Zhang et al., 2015) and Google (Li et al., 2022), creative writing samples from Reddit Writing Prompts (Fan et al., 2018), general educational web articles from FineWeb-EDU (Lozhkov et al., 2024), and news articles from XSum (Narayan et al., 2018) and CNN/DailyMail (See et al., 2017). As a holdout domain to measure out-of-distribution performance, we also include the Enron email dataset (Cohen, 2015).

Then, we generate an AI example corresponding to each human example following the synthetic mirroring procedure introduced in Emi & Spero (2024). We use GPT-4.1, Claude 4 Sonnet, and Gemini 2.5 Flash. We also include Llama-3.3-70B-Instruct-Turbo as a holdout LLM to measure performance on out-of-distribution LLMs. Our final train, test, and val splits contain 60k, 6k, and 2.4k examples respectively. We estimate the cost of creating this dataset to be roughly $530. Additional dataset summary statistics can be found in Tables 11 to 15.

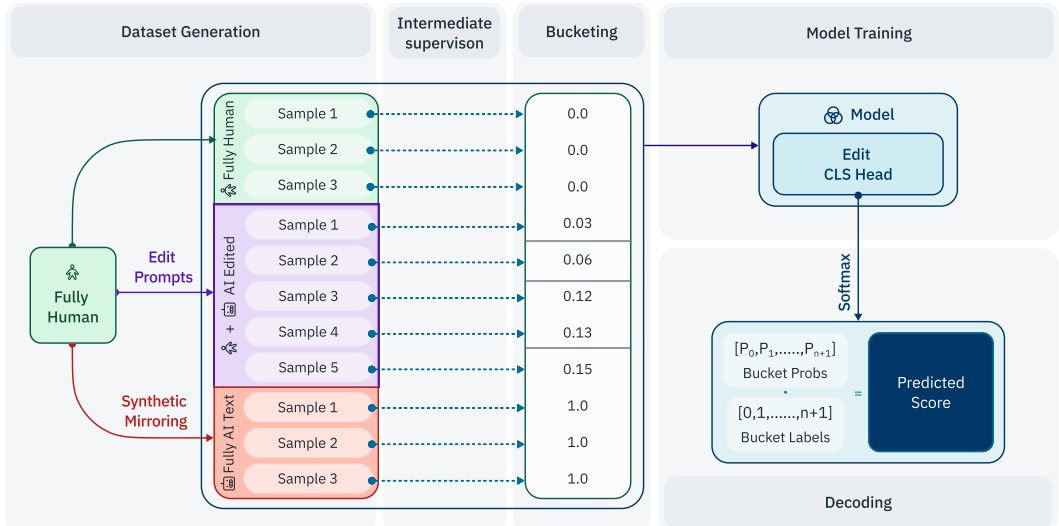

Figure 3: EDITLENS architecture. We generate fully AI and AI-edited versions of human source texts, then use lightweight similarity metrics as intermediate supervision. We partition the texts into $n$ buckets according to supervised score and experiment with training both a regression model and $n$-way classification models, then using weight-average decoding to obtain a numerical score.

## 3.2 EDIT PROMPTS

We collected a set of editing prompts by first prompting ChatGPT 4o, Claude Sonnet 4, and Gemini 2.5 Pro, then adding a small number of prompts written by the authors. In total, we collected 303 editing prompts. The full list of prompts and summary statistics about the categories and contributors can be found in Tables 9 and 10. While this list of prompts is not exhaustive, it encompasses a significant coverage of the different ways that people use AI to edit texts. We split this list of prompts into train, test, and validation splits so that the model cannot overfit to a particular set of prompts.

## 3.3 INTERMEDIATE SUPERVISION METRICS

We experiment with two methods for labeling the "difference" $\Delta(x, y)$ in a text before and after AI editing. The first is the cosine distance (1 - cosine similarity) between the Linq-Embed-Mistral (Choi et al., 2024) embeddings of the source text and the AI-edited version. We chose this embedding due to its strong all-around performance on the MTEB benchmark (Muennighoff et al., 2023).

The second is a precision-based method similar to the embedding-based ROUGE proposed by Ng & Abrecht (2015): given a minimum ($a$) and maximum ($b$) sequence length, we enumerate all phrases (including overlaps) of between $a$ and $b$ words in the source and edited texts. We compute the pairwise cosine similarity between phrases in the source and edited texts, then count the number of phrases in the edited text with a cosine similarity above a threshold $\tau$ for *any* phrase in the source text. This count is divided by the total number of phrases in the edited text, making it a precision-based metric. We refer to this metric as the *soft n-grams* score throughout the paper. Soft n-grams reduces to n-gram overlap between source and target when $\tau = 1$. We choose soft n-grams because it expresses similarity when the AI editor replaces a phrase or word with a semantically similar one rather than requiring exact matching. We note that this supervision metric is shortening-invariant, i.e., simply deleting text from the source still yields a soft n-grams score of 1.

## 3.4 HUMAN AGREEMENT WITH INTERMEDIATE SUPERVISION METRICS

How well do these automatic metrics actually capture the extent of AI editing in a text? To support our choice of intermediate supervision metric, we conduct a study that

asks human annotators to compare two AI-edited versions of the same text after findings by Russell et al. (2025) that humans are effective detectors of AI-authored text.

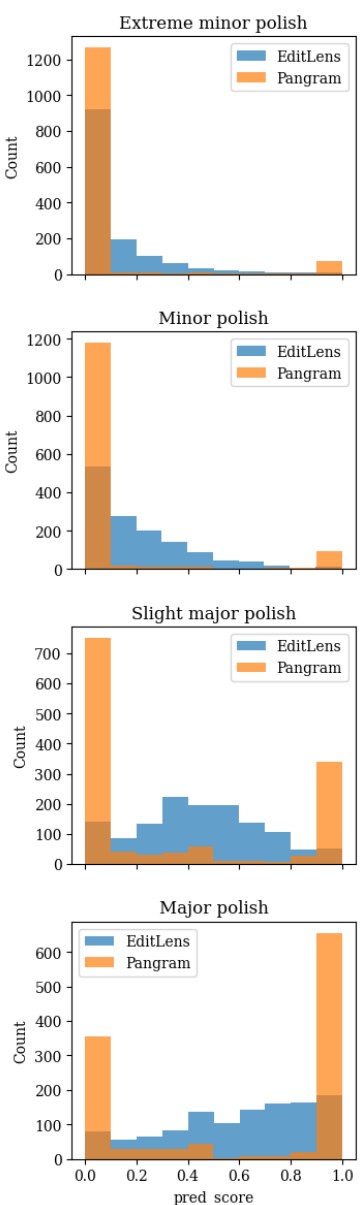

Figure 4: Distributions for ED-ITLENS and Pangram on the AI Polish dataset (Saha & Feizi, 2025). Pangram overwhelmingly tends to predict a score of either 0 or 1, while EDITLENS captures the increasing levels of AI polish applied to the texts.

**Task setup.** Annotators are shown 3 texts side-by-side: a human written *source text* alongside 2 AI-edited versions of the source text. The labeling interface can be seen in Figure 7. Between the two AI texts, annotators are asked to select which text contains more AI edits. Annotators *must* select a text, so no ties are allowed, but they are also asked for their confidence in their choice on a 1 - 5 scale. Annotators have the option to leave freeform comments on each task, but were not required to do so. We recruited 7 annotators with extensive daily exposure to both human writing and AI-generated texts. Each annotator completed all 200 tasks, and was compensated $400 (USD).

**Task generation procedure.** We randomly sample 200 human-written texts of between 50 and 300 words from our test set. We then generate two AI-edited versions of each source text using randomly assigned prompts.

**Agreement with metrics.** We report Krippendorff's $\alpha$ Krippendorff (1980) for 6 annotators[3] and each of our two supervision metrics by treating each metric as an eighth annotator by designating the higher scoring text as the metric's selection. $\alpha$ = 0.72, CI [0.66, 0.77] for cosine and $\alpha$ = 0.71, CI [0.65, 0.76] for soft n-grams (2,000 resampling units).

### 3.5 MODELING DETAILS

Using QLoRA (Dettmers et al., 2023), we finetune models of between 3 and 24B parameters from the Mistral and Llama families. We use QLoRA to sweep the widest possible range of sizes of base models to use as a backbone that fit in VRAM on a single GPU. We leave other finetuning and modeling architecture choices to future work. We experiment with directly training a regression head using MSE loss as well as training an $n$-way classification model, then decoding the output to a score, using weighted-average decoding rather than traditional argmax decoding. Additional modeling details can be found in the Appendix.

## 4 RESULTS

EDITLENS demonstrates significantly more nuanced AI detection than existing classifiers through both quantitative metrics and qualitative analysis. We report results for our best ED-ITLENS model according to ternary classification metrics (see Section 4.3 and Table 2) trained on soft n-grams and cosine scored data after a hyperparameter sweep. Both models have a Mistral Small (24B) backbone and were trained on a 4-way classification task[4].

---

[3]The 7th annotator was excluded because they had poor agreement with all other annotators, but agreement with both metrics was still moderate when including the 6th annotator. Krippendorf's $\alpha$ was 0.66, CI [0.60, 0.71] for cosine score and 0.65, CI [0.59, 0.71] for soft n-grams (2,000 resampling units).

[4]Additional results for different model sizes, model families, and values of $n$ for $n$-way classification can be found in Tables 16-19

We compare EDITLENS with several open- and closed-source AI detection baselines. On the AI Polish dataset (APT-Eval) Saha & Feizi (2025), EDITLENS achieves substantially stronger correlations with edit magnitude metrics compared to binary detectors (correlation 0.606), markedly outperforming the best binary baseline Pangram (correlation 0.491). This quantitative superiority is complemented by clear qualitative differences: while binary classifiers like Pangram predict scores clustered near 0 or 1, EDITLENS produces a nuanced distribution that appropriately tracks increasing levels of AI polish from minor to major edits. The model's regression-based approach enables it to achieve state-of-the-art performance across evaluation paradigms, delivering 94.0% accuracy in binary classification (human vs. any AI) and 90.2% accuracy in ternary classification (human vs. AI-edited vs. AI-generated), substantially outperforming existing binary and ternary detection methods. Additionally, EDITLENS generalizes effectively outside its training distribution: to unseen prompts, LLMs, and domains, to human-edited AI text in the BEEMO dataset, and to AI-edited AI text as well as multi-edited AI text.

## 4.1 AI POLISH DATASET

We first compare the performance on the AI Polish dataset (APT-Eval) of EDITLENS against the best-performing binary AI classifier, Pangram. APT-Eval contains both degree-based AI-edited text, with 4 discrete categories (extreme minor, minor, slight major, and major polish levels), as well as percentage-based AI-edited text, where LLMs were asked to edit a certain percentage of the text, varying from 1-75%.

While there are no direct or exact labels, the score should generally monotonically increase as the amount of requested polish increases. In Figure 4, we qualitatively assess the distribution of the model prediction scores on the degree-based edits. We can see a clear difference between the behavior of EDITLENS versus the behavior of Pangram. Pangram almost always predicts a score very close to 0 or 1, while EDITLENS is able to quantify the increasing levels of polish applied. We show the equivalent distributions for percentage-based polishing in the Appendix.

Quantitatively, we also report the correlation value between the EDITLENS predicted score and the similarity metrics between source and target provided by APT-eval in Table 5. For EDITLENS and all binary classification baselines, we measure the Pearson correlation coefficient ($r$) between the prediction scores and the semantic similarity (-0.606), Levenshtein distance (0.799), and Jaccard distance (0.781) metrics between the pre-AI-polished and post-AI-polished documents. Stronger correlation values mean that the model is able to faithfully track edit magnitude across examples and assign higher scores as semantic similarity decreases (and Levenshtein/Jaccard distances increase), and lower scores when the edited text remains close to the source. EDITLENS exhibits a significant correlation between these similarity metrics and its scores, while the binary AI detectors correlate less strongly with these metrics.

## 4.2 PERFORMANCE AS A BINARY CLASSIFIER

In the binary classification setting, how does EDITLENS treat mixed text? Different use cases may have different standards for what they consider an acceptable amount of AI-generated text–a professor may allow the use of AI assistance for proofreading, but disallow fully AI-generated essays.

To measure the flexibility of our model and the baselines to be able to adjust to different sensitivity levels, we calibrate and compute the performance of each model on two settings: fully human-written vs. any AI-edited or AI-generated text, fully human-written and AI-edited text vs. AI-generated text. We describe the calibration procedure in the Appendix. Model accuracy and F1-scores can be found in Table 1. Notably, EDITLENS outperforms our three binary baselines, FastDetectGPT, Binoculars, and Pangram, on our test set consisting of fully human-written, fully AI-generated, and AI-edited texts.

## 4.3 PERFORMANCE AS A TERNARY CLASSIFIER

To compare with categorical mixed AI detection models, we evaluate each model on three classes: human, AI-generated, or AI-edited. To convert each binary classifier into a ternary classifier, we find two thresholds using the calibration procedure above on a held-out validation set, optimizing the F1

| (a) Human vs. Any AI | | |
|---|---|---|
| **Model** (Threshold) | **Acc. (%)** | **F1** |
| FastDetectGPT $_{0.009}$ | 69.1 | 80.5 |
| Binoculars $_{0.362}$ | 68.6 | 81.4 |
| Pangram $_{0.001}$ | 80.7 | 83.7 |
| EDITLENS (Cosine) $_{0.039}$ | 93.8 | 95.4 |
| **EDITLENS (SNG)** $_{(0.041)}$ | **94.0** | **95.6** |

| (b) Fully AI vs. AI-Edited + Human | | |
|---|---|---|
| **Model** (Threshold) | **Acc. (%)** | **F1** |
| FastDetectGPT $_{0.889}$ | 90.6 | 84.4 |
| Binoculars $_{0.601}$ | 31.4 | 47.7 |
| Pangram $_{0.998}$ | 92.3 | 89.0 |
| EDITLENS (SNG) $_{(0.998)}$ | 94.3 | 90.2 |
| **EDITLENS (Cosine)** $_{(0.960)}$ | **96.4** | **94.1** |

Table 1: Accuracy and F1-score on two binary classification tasks: (a) human vs. any AI generated or edited texts and (b) fully AI-generated texts vs. AI-edited and human texts. Thresholds were calibrated using the val set. "SNG" and "Cosine" denote EDITLENS trained with soft n-grams supervised data and cosine score supervised data, respectively.

score between the human/mixed and mixed/AI classes. The decoding procedures for GPTZero and DetectAIve are detailed in the Appendix.

| Model | Type | Accuracy | Macro-F1 | Human F1 | AI-Gen. F1 | AI-Edited F1 |
|---|---|---|---|---|---|---|
| Binoculars | Binary | 48.5 | 50.1 | 54.7 | 58.4 | 37.4 |
| FastDetectGPT | Binary | 60.6 | 56.8 | 25.1 | 84.4 | 61.0 |
| Pangram | Binary | 73.0 | 69.5 | 76.4 | 89.0 | 43.2 |
| DetectAIve | Ternary | 57.8 | 52.5 | 79.9 | 16.1 | 61.5 |
| GPTZero | Ternary | 74.7 | 72.7 | 77.3 | 89.8 | 50.9 |
| EDITLENS $_{\text{Soft N-Grams}}$ | Regression | 89.7 | 89.9 | 89.6 | 94.1 | 86.1 |
| **EDITLENS $_{\text{Cosine}}$** | **Regression** | **90.2** | **90.4** | **90.4** | **94.1** | **86.8** |

Table 2: Ternary classification performance across different models. Thresholds were calibrated using the validation set. "Soft N-Grams" and "Cosine" denote EDITLENS trained with soft n-grams supervised data and cosine score supervised data, respectively.

### 4.4 NEGATIVE CONTROL: HUMAN-EDITED HUMAN TEXT

In order to ensure that EDITLENS is specifically detecting AI-created edits and not *any* edits, we run a negative control on human-edited human text. We use the ArgRewrite V.2 (Kashefi et al., 2022) dataset, a corpus of annotated argumentative revisions, in order to evaluate the effect on the EDITLENS score of human-editing of human-written text.

We find that the average EDITLENS score (on a 0-1 scale) on human-edited human text is 0.012. The average EDITLENS score of human text 0.009, and the average EDITLENS score of AI-edited human text on the same dataset is 0.86. We conclude from these results that EditLens does not predict human-edited human text as AI-edited. Human-edited human text is predicted effectively the same as human text by EditLens.

This result is due to the fact that human-edited human text is implicitly present in the human-labeled portion of our dataset (e.g. news articles go through multiple rounds of human editing before publication).

### 4.5 POSITIVE CONTROL: AI-EDITED AI TEXT

We also evaluate our detector's mean score difference on AI-edited, AI-generated text, to make sure that this kind of text is always predicted as fully AI-generated. We take synthetic mirrors of our original human dataset, considered to be 'AI-generated documents' and edit them using our held-out prompt set. On a dataset size of n=412, the mean score difference for a single edit pass on an originally human text is 0.38. The mean score difference for a single edit pass on an originally AI text is -0.05.

### 4.6 GENERALIZATION TO HUMAN-EDITED AI TEXT (BEEMO)

While the majority of our studies focus on AI-edited human writing, we also evaluate the performance of EDITLENS on human-edited AI text using the BEEMO (Artemova et al., 2024) dataset, which includes human expert-edited versions of AI model outputs. We find that the model adequately generalizes to human-edited AI text. The average decrease in score from the model output to the human-edited model output is $0.33 \pm 0.30$, with the score decreasing after human-editing in 88.9% of the documents. More details are presented in the Appendix.

### 4.7 CASE STUDY: GRAMMARLY EDIT DATASET

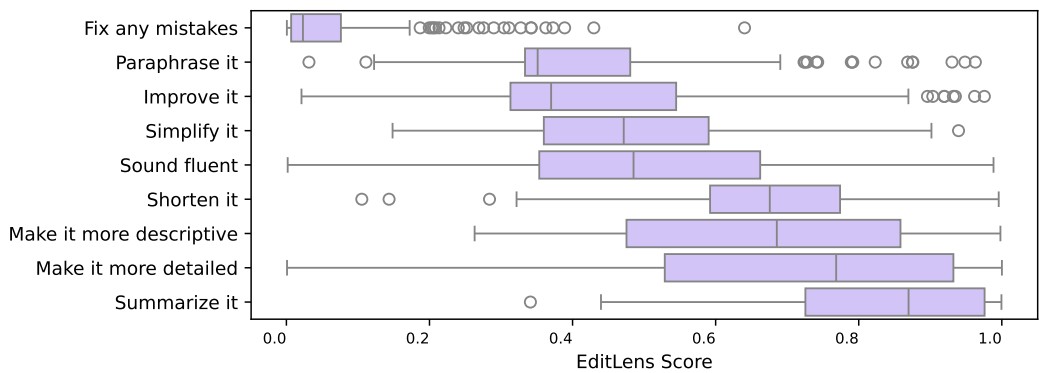

Figure 5: Distribution of EDITLENS scores on dataset from Grammarly by edit instruction.

Grammarly[5] is a popular subscription-based AI writing assistant that allows users to edit text using both pre-filled and custom prompts within a native word processor. We manually collect a dataset of 1768 samples using 9 of the default prompts offered by Grammarly to simulate typical user queries for AI editing by sampling 197[6] human-written source texts and applying each of the 9 edits to them in the Grammarly web interface. In Figure 5, we present the distributions of EDITLENS scores on examples from each editing instruction sorted by the median. Perhaps counterintuitively, EDITLENS considers "Fix any mistakes" the most minor of all edits, while "Summarize this" and "Make it more detailed" are the most invasive edits. In Figures 11 and 10 we show this is also true according to both the cosine and soft n-grams scores of the examples.

## 5 RELATED WORK

**Binary AI-Generated Text Detectors.** Several works have explored the binary setting of distinguishing fully-human from fully-AI-generated text. DetectGPT (Mitchell et al., 2023), FastDetect-GPT (Bao et al., 2024), DNA-GPT (Yang & Cheng, 2024), and Binoculars (Hans et al., 2024) are training-free approaches that leverage statistical properties of AI-generated text to perform binary detection. Ghostbusters (Verma et al., 2024) is an open-weight classifier trained on simple features from the text, while closed-source classifiers such as GPTZero (Tian & Cui, 2023) and Pangram (Emi & Spero, 2024) have more recently emerged as accurate AI text classifiers. All of the above methods operate in the *post-hoc* setting, in contrast to work on watermarking (Kirchenbauer et al., 2024) in which the model's decoding algorithm is modified to enable detection.

**Heterogeneous Mixed Text Detection.** As described above, previous work on mixed AI and human text detection focuses on the heterogeneous case: where distinct boundaries can be drawn between fully AI-generated and fully human-written segments. Examples of these works include AI Boundary Detection with RoFT (Kushnareva et al., 2024), SeqXGPT (Wang et al., 2023), HaCo-Det (Su et al., 2025), and PALD (Lei et al., 2025).

---

[5] https://www.grammarly.com/
[6] Occasionally, Grammarly would abstain, leaving us with fewer than 197 * 9 samples.

**Categorical Mixed Text Detection.** Alternatively, some previous work has instead focused on mixed text as an additional category or categories in addition to human and AI. DetectAIve (Abassy et al., 2024), HERO (Wang et al., 2025), and GPTZero (Tian & Cui, 2023) are all examples where mixed categories have been added. We find the limitation of this approach is that the amount of editing cannot be quantified: all mixed text is treated as the same.

**Human-Edited AI Text.** In addition to our problem setting of AI-edited human written text, there are also studies and datasets focusing on human-edited AI-generated text. Beemo (Artemova et al., 2024) is a benchmark focusing on expert-edited AI text. LAMP (Chakrabarty et al., 2025) is a corpus of LLM-generated paragraphs that have been improved by professional writers according to a defined taxonomy.

**Paraphrasers and Humanizers.** Several previous works have studied the effects of automated paraphrasers (Krishna et al., 2023; Russell et al., 2025; Sadasivan et al., 2025; Cheng et al., 2025) and "humanizers" (Masrour et al., 2025) on how they degrade AI-generated text. We explore the effect of AI rewriting of AI outputs as it relates to our model in the results.

## 6 DISCUSSION AND LIMITATIONS

While EDITLENS can quantify the degree of editing as measured by distance metrics from embeddings commonly used for retrieval, classification, etc., the limitation of EDITLENS is that it cannot say *how* the text was edited or produced. Quantifying edits on a one-dimensional scale is reductive: there are multiple modes of editing such as tone adjustment versus adding original content that are important to differentiate, and EDITLENS treats all types of editing on the same footing. Future work could quantify the extent of editing on these different qualitative axes. Finally, the embeddings themselves could be further optimized to align better with human perception of AI editing or other metrics, such as factual integrity of the original source or semantic similarity, rather than using a simple off-the shelf embedding for retrieval. The embeddings themselves could also be optimized directly for the AI detection task. We leave these experiments for future work.

## 7 CONCLUSION

In this study, we introduce the task of continuous fine-grained AI edit prediction, and show that EDITLENS, based on simple embedding-based supervision on a finetuned language model, significantly outperforms existing AI detection approaches. By moving beyond binary or categorical detection frameworks, our method provides a more nuanced view of mixed-authorship text, quantifying the magnitude of AI editing rather than simply flagging the presence of AI-generated text. This capability enables more flexible policy decisions around the use of generative AI. We release our dataset and models to encourage future research in this area.

## 8 ETHICS STATEMENT

Our research involved using 3 human subjects to annotate the degree of AI-editing present in a text. We obtained informed consent from the subjects and fairly compensated them for their labor. We commit to maintaining their privacy.

Inaccurate AI detection software can cause harm as false accusations of AI misconduct can result in serious consequences, including emotional trauma, reputation damage, and undue punishments for academic misconduct. We acknowledge that our model has a non-zero error rate and its errors may result in such harms. We commit to continuing to engage with the academic community to educate others on appropriately contextualizing and communicating the results of AI detection software. We also commit to releasing the model for non-commercial use only and responsibly vetting access to researchers and educators.

We intend for our contribution to the research on AI detection to ultimately mitigate harm by providing a more nuanced picture of AI usage than binary AI detection classifiers. The ability to calibrate the sensitivity level of the regression model is also a step towards mitigating the false positive rate and lowering the overall number of false accusations of AI misconduct.

## 9 REPRODUCIBILITY STATEMENT

We release the dataset, source code, and model weights at `https://github.com/pangramlabs/EditLens`.

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

## A  Differences with the Heterogeneous Mixed Text Detection Task

The PaLD (Lei et al., 2025) formulation considers a text $x$ segmented as $x = x_1 \cdots x_n$, where each segment $x_i$ is assumed to originate from either a human or an LLM, i.e., $x_i \sim P_{\text{human}}$ or $x_i \sim P_{\text{LLM}}$. The learning objective is to infer latent per-segment authorship labels $a_{1:n} \in \{\text{human}, \text{LLM}\}^n$ (and optionally segment boundaries), estimating $p_\theta(a_{1:n} \mid x)$ and predicting $\hat{a}_{1:n} = \arg\max p_\theta(a_{1:n} \mid x)$. In contrast, our *homogeneous mixed text prediction* task dispenses with provenance as supervision and regresses an authorship-agnostic *edit magnitude* aligned to a similarity metric. Given pre/post pairs only to derive targets, inference relies on a *single-input* predictor $f_\theta^{\text{ssi}}(y)$ that maps the edited text $y$ directly to $\hat{\Delta}(y) \in [0, 1]$, without segment labels, boundary inference, or reconstruction of the source. This reframing changes (i) the **assumptions** (binary authorship mixture vs. latent, entangled edits), (ii) the **outputs** (label sequence $a_{1:n}$ vs. scalar/regional magnitudes $\Delta$), (iii) the **supervision** (segment-level authorship vs. metric-aligned change signals), and (iv) the **evaluation** (classification metrics such as accuracy/F1 vs. correlation and error against $\Delta$, plus calibration).

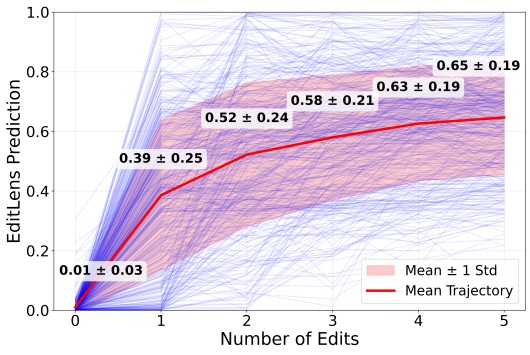

Figure 6: "Trajectory" of EDITLENS scores after subsequent AI edits to a single text.

## B  Performance on multi-edited AI text

We also examine the case where multiple AI-edits have been applied to a single piece of text. We test our model on this case by applying a series of 5 edits to a piece of human-written text and measuring the EDITLENS score after each subsequent edit. In Figure 6, we show that for each edit, the mean score increases monotonically.

## C  Out-of-Domain Performance

During dataset creation, we hold out both a model and a domain to test the ability of our model to generalize to out-of-distribution texts. We created an OOD model test set of 3k examples with

|  | Raw scores | 4 buckets | 5 buckets | 6 buckets |
|---|---|---|---|---|
| **Cosine Distance** | $0.67 \pm 0.06$ | $0.50 \pm 0.05$ | $0.52 \pm 0.05$ | $0.55 \pm 0.05$ |
| **Soft N-Grams** | $0.66 \pm 0.05$ | $0.48 \pm 0.05$ | $0.50 \pm 0.05$ | $0.52 \pm 0.05$ |

Table 3: Agreement (Krippendorff's $\alpha$ with bootstrap SE) between human annotators and proposed intermediate supervision metrics under different bucketing schemes for scores.

Llama-3.3-70B-Instruct-Turbo generated and edited texts as well as an OOD domain test set using the Enron email dataset (Cohen, 2015) as source texts, and measure the degradation in macro-F1 score of our best model, EDITLENS with cosine supervision. On the OOD domain dataset, macro-F1 on the ternary classification task decreases from 0.904 to 0.866 (-0.038). On the OOD LLM dataset, macro-F1 on the ternary classification task decreases from 0.904 to 0.850 (-0.054).

## D    HUMAN AGREEMENT WITH INTERMEDIATE SUPERVISION METRICS

We compute the score for each pair of source and AI-edited texts, then assign each AI-edited text to one of $n$ buckets according to the bucketing scheme described in Section G. All $\alpha$ values are reported in Table 3.

## E    ABLATION: USING A DIFFERENT EMBEDDING MODEL FOR SUPERVISION

We ran an additional ablation study with Qwen3 (Zhang et al., 2025) embeddings as supervision.

BINARY CLASSIFICATION

- **Best Human F1:** 0.96
- **Best AI F1:** 0.93

TERNARY CLASSIFICATION

- **Accuracy:** 0.902
- **Macro F1:** 0.903

| Class | F1 Score |
|---|---|
| Human | 0.891 |
| AI | 0.959 |
| AI-Edited | 0.861 |

Table 4: Per-class F1 scores for ternary classification.

We observe no significant difference in results between models trained on cosine distance of different embeddings.

## F    CALIBRATION PROCEDURE

We calibrate the threshold for the model for both the binary and ternary settings. We first segregate the data into three categories: human, AI, or mixed. Mixed text is human text that has any amount of AI editing applied. Our goal is then to find two thresholds, a lower threshold and an upper threshold. The lower threshold separates the [Human] class from the union of [AI, Mixed] classes. The upper threshold separates the [AI] class from the union of [Human, Mixed] classes. Both thresholds are found by maximizing the F1 score on a separate held out validation set for these two labeling schemas.

# G   MORE MODELING DETAILS

We use QLoRA to sweep both Llama and Mistral families of backbones between 3B and 24B parameters. We experiment with both a direct regression head and a N-way classification head with weighted-average decoding.

**Determining thresholds for fully human and fully AI texts**   Some edits are too small to be detectable, such as adding a single comma, correcting a typo, etc. We choose a minimum threshold of 0.03 for cosine distance threshold and 0.06 for soft n-grams in order to supervise it as AI-edited, chosen through manual inspection and validation of edits we would consider small enough such that the authorship is still entirely human.

Additionally, there are cases on the other end of the spectrum where AI was so pervasive in a text that it essentially rewrote the entire document and it became fully AI-generated. To measure the upper threshold where we would consider a text fully AI-generated, we analyzed the similarity metrics between the sources and their corresponding fully AI-generated synthetic mirrors. We selected thresholds that best separate fully AI-generated synthetic mirrors from the heaviest AI-edited text, which were 0.15 for cosine distance and 0.72 for soft n-grams.

## G.1   REGRESSION FORMULATION

Let $s$ denote the raw similarity score and $\tau_{\text{low}}$ and $\tau_{\text{high}}$ be the low and high thresholds, respectively. We define the scaled similarity score as:

$$\tilde{s} = \begin{cases} 0.0 & \text{if } s \leq \tau_{\text{low}} \\ 1.0 & \text{if } s \geq \tau_{\text{high}} \\ \frac{s - \tau_{\text{low}}}{\tau_{\text{high}} - \tau_{\text{low}}} & \text{otherwise} \end{cases} \tag{1}$$

The regression model directly predicts the scaled similarity score $\hat{s}$ using a mean squared error loss:

$$\mathcal{L}_{\text{MSE}} = \frac{1}{n} \sum_{i=1}^{n} (\tilde{s}_i - \hat{s}_i)^2 \tag{2}$$

where $n$ is the number of training examples.

## G.2   CLASSIFICATION FORMULATION

For the classification approach, we discretize the similarity scores into $N$ buckets, where $N \in \{4, 5, 6\}$. Given minimum and maximum thresholds $\tau_{\text{min}}$ and $\tau_{\text{max}}$, we define the bucket assignment function:

$$b(s) = \min\left(N - 1, \left\lfloor \frac{s - \tau_{\text{min}}}{\tau_{\text{max}} - \tau_{\text{min}}} \cdot N \right\rfloor\right) \tag{3}$$

The midpoint of bucket $j$ is given by:

$$m_j = \tau_{\text{min}} + \frac{(j + 0.5) \cdot (\tau_{\text{max}} - \tau_{\text{min}})}{N} \tag{4}$$

We train the classification model using cross-entropy loss:

$$\mathcal{L}_{\text{CE}} = -\frac{1}{n} \sum_{i=1}^{n} \log p(b(s_i)|x_i) \tag{5}$$

where $p(j|x_i)$ is the predicted probability for bucket $j$ given input $x_i$.

During inference, we decode the final similarity score using a weighted average strategy:

$$\hat{s} = \sum_{j=0}^{N-1} p(j|x) \cdot m_j \tag{6}$$

where $p(j|x)$ is the predicted probability for bucket $j$ and $m_j$ is the corresponding bucket midpoint.

### G.3 ARCHITECTURE AND OPTIMIZATION

We train the model for 1 epoch with AdamW using a batch size of 24 and a constant learning rate of 3e-5. We initialize the model with pretrained weights from the base model and target *all* linear layers: self-attention QKV, output, and all linear layers in the MLP. We use a LayerNorm and single linear layer as the head for both prompt classification and edit heads and supervise both jointly in a multi-task learning routine. On 8 A100 GPUs, this takes approximately 8 hours for the largest model.

## H TERNARY CLASSIFIER DECODING

GPTZero reports probabilities of three classes: "human", "AI", and "mixed," so we simply use argmax decoding to select the highest probability class. DetectAIve reports probabilities of four classes: "human", "AI", "AI Polished", and "AI humanized". We attempted to group "AI humanized" predictions with both the "AI" and the "AI Polished" categories for ternary classification, and found that grouping with "AI" produced a higher F1 score. Therefore, we group "AI Humanized" and "AI" into a single category for purposes of comparison.

## I TERNARY CLASSIFICATION CONFUSION MATRICES

Analyzing the confusion matrix, we see that EDITLENS exhibits much stronger performance on the AI-edited text category than the strongest ternary classifier, GPTZero. While both EDITLENS and GPTZero are nearly perfect at distinguishing fully AI-generated text from fully human-written text, EDITLENS is the only model able to also consistently detect AI-edited text as a distinct category from fully human and fully AI.

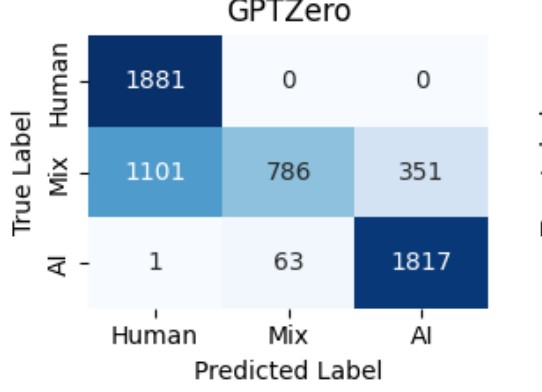
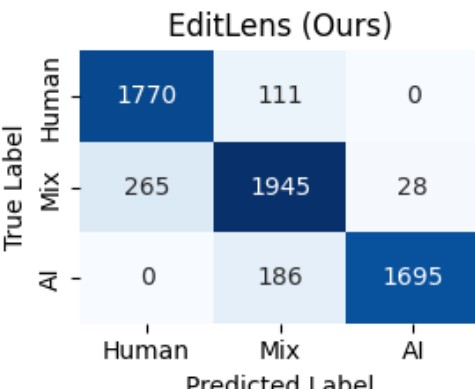

| Model | $r$, semantic sim. ↓ | $r$, Levenshtein ↑ | $r$, Jaccard ↑ |
|---|---|---|---|
| FastDetectGPT | -0.287 | 0.331 | 0.280 |
| Binoculars | -0.291 | 0.326 | 0.274 |
| Pangram | -0.491 | 0.615 | 0.556 |
| **EDITLENS (Ours)** | **-0.606** | **0.799** | **0.781** |

Table 5: Pearson correlation coefficients by model.

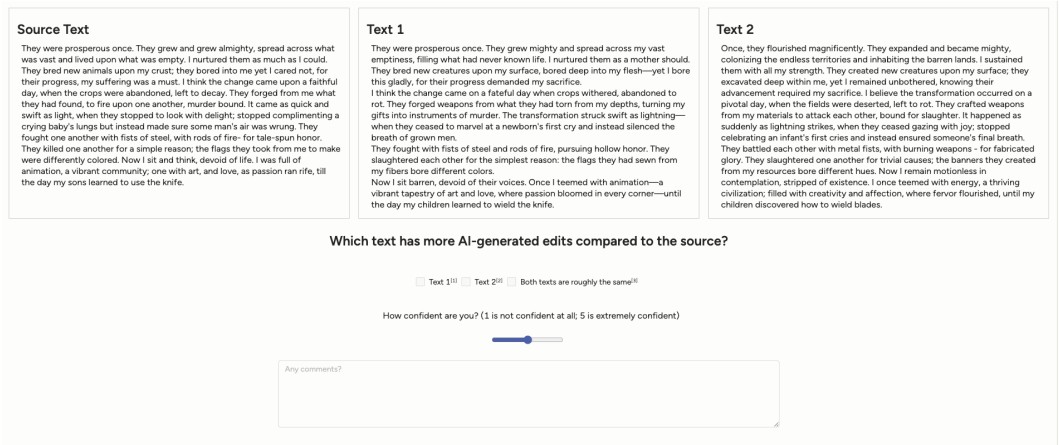

Figure 7: A screenshot of our data annotation interface.

## J    CORRELATION BETWEEN EDITLENS PREDICTIONS AND AI POLISH SIMILARITY METRICS

## K    DATA GENERATION MODELS

| Checkpoint Name | API Provider | # Params | Open Source |
|---|---|---|---|
| `claude-sonnet-4-20250514` Anthropic (2025) | Anthropic API | ? | No |
| `meta-llama/Llama-3.3-70B-Instruct-Turbo` Meta (2024c) | Together.ai API | 70B | Yes |
| `gpt-4.1-2025-04-14` OpenAI (2025) | OpenAI API | ? | No |
| `gemini-2.5-flash` Google (2025) | Gemini API | ? | No |

Table 6: Models used for dataset generation

## L    EMBEDDING MODELS

| Checkpoint Name | # Params | Dim. Size |
|---|---|---|
| `Linq-AI-Research/Linq-Embed-Mistral` Choi et al. (2024) | 7B | 4096 |
| `sentence-transformers/all-MiniLM-L6-v2` Reimers et al. (2021) | 22.7M | 384 |

Table 7: Embedding models used for supervision

## M  BASE MODELS

| Checkpoint Name | # Params | Open Source |
|---|---|---|
| `mistralai/Mistral-Nemo-Base-2407` Mistral (2024) | 12B | Yes |
| `mistralai/Mistral-Small-24B-Base-2501` Mistral (2025) | 24B | Yes |
| `meta-llama/Llama-3.1-8B` Meta (2024a) | 8B | Yes |
| `meta-llama/Llama-3.2-3B` Meta (2024b) | 3B | Yes |

Table 8: Base models used for training

## N  MORE RESULTS ON HUMAN-EDITED AI TEXT

We focus on the human-edited AI versions of "Generation" and "OpenQA" categories of BEEMO, because the other categories, such as "Rewrite" and "Summarize", are already themselves AI-edited versions of human text, "Closed QA" the answers are so tightly constrained we would consider the answers to be human-written, and we would not consider the model outputs fully AI-generated. We also measure the correlation coefficient between our similarity metrics and the model scores. The intuition for this is that if the human edit is more invasive, we would expect the similarity metrics to increase, and the model score to decrease. As expected, we find a moderate negative correlation between our model's scores and the similarity, with $-0.396$ for cosine distance and $-0.501$ for soft n-grams.

In Figures 8 and 9, we present the output distribution of EDITLENS for BEEMO's Generation and OpenQA splits, on the fully AI-generated text (orange) and human-edited version (blue).

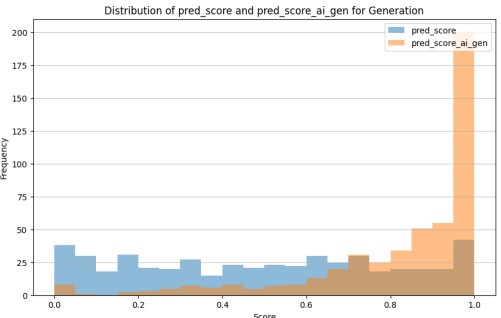

Figure 8: BEEMO Generation    Figure 9: BEEMO OpenQA

As is shown in the figures, the predicted score distribution moves significantly towards human-generated following editing, as expected.

## O  EDITING PROMPTS

Table 9: Full list of editing prompts by split and contributor.

| TRAIN | |
|---|---|
| **Editing Prompt** | **Contributor** |
| Change this so it fits what a business person would want | ChatGPT 4o |
| Make this sound more sure and strong | ChatGPT 4o |
| Edit this for people who don't know the topic well | ChatGPT 4o |

| Editing Prompt | Contributor |
| --- | --- |
| Make this more direct and bold | ChatGPT 4o |
| Make this sound fair and not take sides | ChatGPT 4o |
| Make this sound more serious and important | ChatGPT 4o |
| Make this sound more excited and energetic | ChatGPT 4o |
| Make this easier for someone who doesn't know that much about it | ChatGPT 4o |
| Make this more accessible to a non-expert reader | ChatGPT 4o |
| Adapt this for readers with no prior background in the topic | ChatGPT 4o |
| Write this like you're talking to someone | ChatGPT 4o |
| Make this sound more doubtful and questioning | ChatGPT 4o |
| Adjust the voice to sound more academic. | ChatGPT 4o |
| Make this sound more relaxed and friendly | ChatGPT 4o |
| Write this in a way that top company leaders would like | ChatGPT 4o |
| Make this more formal and proper | ChatGPT 4o |
| Make this easier to remember and repeat | ChatGPT 4o |
| Tailor this message to suit a lay audience | ChatGPT 4o |
| Make this sound more exciting and well written | ChatGPT 4o |
| Make this sound more serious and proper | ChatGPT 4o |
| Use a smart and serious tone like in official stuff | ChatGPT 4o |
| Edit this to sound more urgent and important | ChatGPT 4o |
| Make this more convincing and easier to understand | ChatGPT 4o |
| Change this so it's easy for a teen to read | ChatGPT 4o |
| Make this more logical and fact-based | ChatGPT 4o |
| Make the consequences feel more important | ChatGPT 4o |
| Make this sound uplifting and encouraging | ChatGPT 4o |
| Rewrite this to align with a formal tone | ChatGPT 4o |
| Make this more convincing and clear | ChatGPT 4o |
| Change this so a 5th grader can understand it | ChatGPT 4o |
| Take out hard words and explain them in a simple way | ChatGPT 4o |
| Fix this to make it more interesting | ChatGPT 4o |
| Change this to make it as strong as possible | ChatGPT 4o |
| Rewrite this to better suit a business audience | ChatGPT 4o |
| Use simpler words so anyone can understand this | ChatGPT 4o |
| Make this sound more like school writing | ChatGPT 4o |
| Make this sound nicer and more fun | ChatGPT 4o |
| Change this to sound more thoughtful | ChatGPT 4o |
| Make this funnier and more lighthearted | ChatGPT 4o |
| Use better words to make this sound smarter | ChatGPT 4o |
| Add some personality to this | ChatGPT 4o |
| Make this sound more like professional writing | ChatGPT 4o |
| Give examples to help make this clearer | ChatGPT 4o |
| Edit this with clear examples to help explain this better | ChatGPT 4o |
| Add details that create a mood or feeling | ChatGPT 4o |
| Tell some of the story behind this to help understand it | ChatGPT 4o |
| Explain more about what this means and why it matters | ChatGPT 4o |
| Explain the main points using real examples | ChatGPT 4o |
| Add descriptions of sounds, smells, textures, and how things feel | ChatGPT 4o |
| Include exact sizes, colors, and what things look like | ChatGPT 4o |
| Add details about the setting and background | ChatGPT 4o |
| Add details that help readers see, hear, and feel what's happening | ChatGPT 4o |
| Add conversations and quotes to make scenes more real | ChatGPT 4o |
| Develop this text further by explaining the implications | ChatGPT 4o |
| Help readers picture this more clearly with specific details | ChatGPT 4o |
| Describe how things look, sound, or feel more | ChatGPT 4o |
| Use more interesting and specific describing words | ChatGPT 4o |

| Editing Prompt | Contributor |
| --- | --- |
| Add comparisons to help explain things better | ChatGPT 4o |
| Use real-life examples to show what you mean | ChatGPT 4o |
| Add background facts to back up these points | ChatGPT 4o |
| Add true stories or examples from real life | ChatGPT 4o |
| Use specific names of places, brands, and things | ChatGPT 4o |
| Add more ideas or facts that support what you're saying | ChatGPT 4o |
| Add details about the place and situation | ChatGPT 4o |
| Share what experts think or what research shows | ChatGPT 4o |
| Use real examples to explain this better | ChatGPT 4o |
| Use stronger, more exciting action words | ChatGPT 4o |
| Add illustrative examples to clarify these points | ChatGPT 4o |
| Ensure the sentences transition smoothly from one idea to the next | ChatGPT 4o |
| Help the ideas move from one to the next easily | ChatGPT 4o |
| Make the ideas connect better | ChatGPT 4o |
| Make the language flow more fluidly without sounding forced | ChatGPT 4o |
| Can you fix parts that sound choppy or off? | ChatGPT 4o |
| Make this writing smoother and better | ChatGPT 4o |
| Make the words fit together better | ChatGPT 4o |
| Make this easier and smoother to read | ChatGPT 4o |
| Help the ideas connect more smoothly | ChatGPT 4o |
| Make this sound like someone who speaks English well wrote it | ChatGPT 4o |
| Make the sentences flow better | ChatGPT 4o |
| Make the rhythm of the sentences better | ChatGPT 4o |
| Remove all extra words and phrases | ChatGPT 4o |
| Trim this down while keeping the tone and meaning intact | ChatGPT 4o |
| Remove extra or repeated words | ChatGPT 4o |
| Use clearer words but keep the meaning | ChatGPT 4o |
| Get rid of long or confusing parts | ChatGPT 4o |
| Make this shorter and more direct | ChatGPT 4o |
| Take out anything extra but keep the good parts | ChatGPT 4o |
| Say this in fewer words but still make it strong | ChatGPT 4o |
| Take out words that aren't needed to make this better | ChatGPT 4o |
| Take out words that don't add anything | ChatGPT 4o |
| Cut this down but keep the same meaning and style | ChatGPT 4o |
| Rearrange this content for a clearer argument progression | ChatGPT 4o |
| Put this in a better order | ChatGPT 4o |
| Put similar ideas together more clearly | ChatGPT 4o |
| Build up to a strong ending | ChatGPT 4o |
| Put this in an order that makes more sense | ChatGPT 4o |
| Set this up in a way that's easier to read | ChatGPT 4o |
| Group ideas that go together and add connections | ChatGPT 4o |
| Move things around to make the main points stand out | ChatGPT 4o |
| Split this into paragraphs that connect better | ChatGPT 4o |
| Make sentences match and sound good together | ChatGPT 4o |
| Put the most important stuff first | ChatGPT 4o |
| Organize this information in a more reader-friendly format | ChatGPT 4o |
| Tell the story in a more interesting way | ChatGPT 4o |
| Make this work better overall | ChatGPT 4o |
| Make this writing more polished and effective | ChatGPT 4o |
| Refine this text to improve its overall impact | ChatGPT 4o |
| Fix this but keep the same meaning | ChatGPT 4o |
| Fix this but keep the main point the same | ChatGPT 4o |
| Improve this passage while preserving its original intent | ChatGPT 4o |
| Restate this using different vocabulary and sentence structure | ChatGPT 4o |

| Editing Prompt | Contributor |
|---|---|
| Use easier words that mean the same thing | ChatGPT 4o |
| Share this idea from a new point of view | ChatGPT 4o |
| Say this in a new way | ChatGPT 4o |
| Use different words to say the same thing | ChatGPT 4o |
| Say this using different words and sentence types | ChatGPT 4o |
| Say this using different words and ideas | ChatGPT 4o |
| Say this in a new and interesting way | ChatGPT 4o |
| Paraphrase this to make it simpler and easier to understand | ChatGPT 4o |
| Explain any hard words so people know what they mean | ChatGPT 4o |
| Fix parts that sound weird or hard to read | ChatGPT 4o |
| Say clearly who or what each word is talking about | ChatGPT 4o |
| Make this clear and easy to read | ChatGPT 4o |
| Make sure one idea leads to the next clearly | ChatGPT 4o |
| Make it obvious what causes what | ChatGPT 4o |
| Say this in a clear and simple way | ChatGPT 4o |
| Correct any grammar, punctuation, or spelling errors in this text | ChatGPT 4o |
| Use better words and fix how the sentences are written | ChatGPT 4o |
| Make this more direct and confrontational | Claude Sonnet 4 |
| Make this more memorable and quotable | Claude Sonnet 4 |
| Make this sound more diplomatic and tactful | Claude Sonnet 4 |
| Make this more emotionally resonant | Claude Sonnet 4 |
| Make this more formal | Claude Sonnet 4 |
| Simplify for customers with no technical background | Claude Sonnet 4 |
| Make this suitable for social media sharing | Claude Sonnet 4 |
| Inject enthusiasm and energy into this writing | Claude Sonnet 4 |
| Translate this for a teenage audience | Claude Sonnet 4 |
| Soften the tone while maintaining the message | Claude Sonnet 4 |
| Make this more relatable to the reader's experience | Claude Sonnet 4 |
| Make this more inspiring and motivational | Claude Sonnet 4 |
| Add gravitas and weight to this statement | Claude Sonnet 4 |
| Adopt a more skeptical and questioning tone | Claude Sonnet 4 |
| Make this more casual | Claude Sonnet 4 |
| Adjust for a peer-reviewed academic journal | Claude Sonnet 4 |
| Convert to a more analytical and logical approach | Claude Sonnet 4 |
| Make this appropriate for C-suite executives | Claude Sonnet 4 |
| Add vivid imagery and sensory details to bring this to life | Claude Sonnet 4 |
| Add backstory or context to enrich understanding | Claude Sonnet 4 |
| Add depth and context to make this more comprehensive | Claude Sonnet 4 |
| Include specific measurements, colors, and physical characteristics | Claude Sonnet 4 |
| Flesh out these ideas with supporting information | Claude Sonnet 4 |
| Provide concrete examples to illustrate these points | Claude Sonnet 4 |
| Add sensory details to make this more vivid | Claude Sonnet 4 |
| Use more precise and colorful adjectives | Claude Sonnet 4 |
| Add dialogue and quoted speech to make scenes more vivid | Claude Sonnet 4 |
| Elaborate on the key points with concrete details | Claude Sonnet 4 |
| Add storytelling elements to increase engagement | Claude Sonnet 4 |
| Add descriptive metaphors and similes to enhance understanding | Claude Sonnet 4 |
| Expand with real-world applications | Claude Sonnet 4 |
| Use more evocative and powerful verbs | Claude Sonnet 4 |
| Include expert opinions or research findings | Claude Sonnet 4 |
| Incorporate specific brand names, locations, and proper nouns | Claude Sonnet 4 |
| Paint a clearer picture with specific visual descriptions | Claude Sonnet 4 |
| Use figurative language to make concepts more tangible | Claude Sonnet 4 |
| Include personal anecdotes or case studies | Claude Sonnet 4 |

| Editing Prompt | Contributor |
| --- | --- |
| Make this read like it was written by a native speaker | Claude Sonnet 4 |
| Make this sound more conversational and engaging | Claude Sonnet 4 |
| Improve the rhythm and readability of this writing | Claude Sonnet 4 |
| Make the progression of ideas feel effortless | Claude Sonnet 4 |
| Create smoother connections between these ideas | Claude Sonnet 4 |
| Improve the natural rhythm of this text | Claude Sonnet 4 |
| Smooth out the awkward phrasing in this passage | Claude Sonnet 4 |
| Eliminate any choppy or awkward sentences | Claude Sonnet 4 |
| Remove every unnecessary word and phrase | Claude Sonnet 4 |
| Trim the fat without losing the muscle | Claude Sonnet 4 |
| Make this more concise without losing important information | Claude Sonnet 4 |
| Eliminate wordy expressions and redundancies | Claude Sonnet 4 |
| Tighten this writing by removing unnecessary words | Claude Sonnet 4 |
| Arrange these points in order of importance | Claude Sonnet 4 |
| Create better section breaks and headers | Claude Sonnet 4 |
| Create a more compelling narrative arc | Claude Sonnet 4 |
| Build toward a stronger climax or conclusion | Claude Sonnet 4 |
| Reorganize this for better logical flow | Claude Sonnet 4 |
| Use parallel structure to enhance readability | Claude Sonnet 4 |
| Break this into clearer paragraphs with smooth transitions | Claude Sonnet 4 |
| Revise this to make it more engaging | Claude Sonnet 4 |
| Enhance the overall effectiveness of this passage | Claude Sonnet 4 |
| Refine this writing to make it more professional | Claude Sonnet 4 |
| Enhance this text while maintaining the original meaning | Claude Sonnet 4 |
| Optimize this text for maximum impact | Claude Sonnet 4 |
| Strengthen this writing by improving word choice and structure | Claude Sonnet 4 |
| Transform this into more compelling prose | Claude Sonnet 4 |
| Polish this text for clarity and readability | Claude Sonnet 4 |
| Upgrade the sophistication of this writing | Claude Sonnet 4 |
| Recast these ideas in a different style | Claude Sonnet 4 |
| Rephrase this text to avoid repetition | Claude Sonnet 4 |
| Express these ideas using alternative phrasing | Claude Sonnet 4 |
| Say the same thing but in a fresh way | Claude Sonnet 4 |
| Present the same information from a fresh angle | Claude Sonnet 4 |
| Reframe this argument using different terminology | Claude Sonnet 4 |
| Add precise measurements and timeframes | Claude Sonnet 4 |
| Define any terms that might be unclear | Claude Sonnet 4 |
| Eliminate any ambiguous or vague language | Claude Sonnet 4 |
| Make the cause-and-effect relationships clearer | Claude Sonnet 4 |
| Inject more personality and warmth into this text | Gemini 2.5 Pro |
| Adjust the tone to be more persuasive and convincing | Gemini 2.5 Pro |
| Make this sound more professional and authoritative | Gemini 2.5 Pro |
| Rewrite this to be more empathetic and understanding | Gemini 2.5 Pro |
| Make this sound more urgent and compelling | Gemini 2.5 Pro |
| Make this sound more objective and unbiased | Gemini 2.5 Pro |
| Adopt a more academic and scholarly tone | Gemini 2.5 Pro |
| Improve the transitions between the paragraphs | Gemini 2.5 Pro |
| Create a more effective and engaging opening | Gemini 2.5 Pro |
| Clarify the main idea | Gemini 2.5 Pro |
| Remove any filler words | Gemini 2.5 Pro |
| Remove any jargon or technical terms and explain them in plain language | Gemini 2.5 Pro |
| Replace complex words with simpler alternatives | Gemini 2.5 Pro |
| Identify and eliminate any redundant phrases or words | Gemini 2.5 Pro |
| Make this more concrete and less abstract | Gemini 2.5 Pro |

| Editing Prompt | Contributor |
|---|---|
| Simplify this text for a 5th-grade reading level | Gemini 2.5 Pro |
| Group related ideas together more effectively | Gemini 2.5 Pro |
| Ensure a clear introduction, body, and conclusion | Gemini 2.5 Pro |
| Fix any grammatical mistakes in this text | Gemini 2.5 Pro |
| Write this in a way that a business person would get it | Human |
| Edit this to sound more polite | Human |
| Make this more descriptive | Human |
| Make this more detailed | Human |
| Please add more details to make my argument better | Human |
| Rearrange this | Human |
| Can you make this sound fluent? | Human |
| Rewrite to be more concise and powerful | Human |
| Can you help my essay get a better grade? | Human |
| Rewrite this so it sounds good | Human |
| Make my essay better | Human |
| Make this essay look better | Human |
| Can you improve this? | Human |
| Make this an A paper | Human |
| Remake all of this in a different way | Human |
| Rewrite all of this in different words | Human |
| Can you fix the problems with my argument? | Human |
| Emphasize the key points | Human |
| Make my grammar sound better | Human |

| VAL | |
|---|---|
| Make this connect better with people's feelings | ChatGPT 4o |
| Make this sound more like a friendly conversation | ChatGPT 4o |
| Make this easier for readers to relate to | ChatGPT 4o |
| Make this easier for someone new to the topic to get | ChatGPT 4o |
| Make this sound more businesslike and serious | ChatGPT 4o |
| Change how the sentences are written | ChatGPT 4o |
| Say this in another way | ChatGPT 4o |
| Include contextual information to support these claims | ChatGPT 4o |
| Give more background so it's easier to understand | ChatGPT 4o |
| Use creative comparisons to make ideas clearer | ChatGPT 4o |
| Make this easier to understand | ChatGPT 4o |
| Use simpler language anyone can get | ChatGPT 4o |
| Add better breaks and section titles | ChatGPT 4o |
| Help the paragraphs connect better | ChatGPT 4o |
| Remove anything confusing or unclear | ChatGPT 4o |
| Make this sound more confident and authoritative | Claude Sonnet 4 |
| Adapt this for international readers | Claude Sonnet 4 |
| Translate this into more accessible language | Claude Sonnet 4 |
| Find alternative ways to express these concepts | Claude Sonnet 4 |
| Rework this using varied sentence structures | Claude Sonnet 4 |
| Add atmospheric details to create mood and setting | Claude Sonnet 4 |
| Add environmental and contextual descriptions | Claude Sonnet 4 |
| Restructure this to emphasize the main points | Claude Sonnet 4 |
| Connect these thoughts more seamlessly | Claude Sonnet 4 |
| Elevate the quality of this writing | Claude Sonnet 4 |
| Rewrite this in a more conversational and approachable style | Gemini 2.5 Pro |
| Rewrite this to be more direct and to the point | Gemini 2.5 Pro |
| Edit this into a blog post I can share online | Human |
| Rewrite all of this | Human |

| Editing Prompt | Contributor |
|---|---|
| Proofread this for spelling and grammar errors | Human |

| TEST | |
|---|---|
| Make this sound more formal and school-like | ChatGPT 4o |
| Make this nicer but keep the main point | ChatGPT 4o |
| Change this so people from other countries can get it too | ChatGPT 4o |
| Add a short story to make this more interesting | ChatGPT 4o |
| Show how this works in real life | ChatGPT 4o |
| Say this in fewer words without losing meaning | ChatGPT 4o |
| Edit this to be punchier and more direct | ChatGPT 4o |
| Improve how the sentences sound together | ChatGPT 4o |
| Refine the pacing and cadence of this paragraph | ChatGPT 4o |
| Write a better and more interesting beginning | ChatGPT 4o |
| Make the main idea clearer | ChatGPT 4o |
| Improve this to make it stronger and clearer | ChatGPT 4o |
| Reword this to improve clarity while keeping the meaning | ChatGPT 4o |
| Make sure there's a beginning, middle, and end | ChatGPT 4o |
| Group related ideas and use transitions to improve structure | ChatGPT 4o |
| Adjust the tone to be more friendly | Claude Sonnet 4 |
| Increase the emotional stakes | Claude Sonnet 4 |
| Expand this text with more specific examples and details | Claude Sonnet 4 |
| Show rather than tell by adding scene-setting details | Claude Sonnet 4 |
| Include sounds, smells, textures, and other sensory elements | Claude Sonnet 4 |
| Reduce wordiness while amplifying impact | Claude Sonnet 4 |
| Make this text flow more naturally | Claude Sonnet 4 |
| Rewrite this in different words while keeping the same meaning | Claude Sonnet 4 |
| Lighten the tone and add a touch of humor | Gemini 2.5 Pro |
| Make this longer with more evidence | Human |
| Simplify this text | Human |
| Make this more specific | Human |
| Write this in a way that my teacher would get it | Human |
| Can you make my paper more persuasive? | Human |
| Paraphrase this | Human |
| Can you fix any spelling, grammar, or punctuation issues? | Human |

| | Train | Val | Test | All |
|---|---|---|---|---|
| **Prompt Contributors** | | | | |
| ChatGPT 4o | 52.9% (128) | 50.0% (15) | 48.4% (15) | 52.1% (158) |
| Claude Sonnet 4 | 31.4% (76) | 33.3% (10) | 25.8% (8) | 31.0% (94) |
| Human | 7.9% (19) | 10.0% (3) | 22.6% (7) | 9.6% (29) |
| Gemini 2.5 Pro | 7.9% (19) | 6.7% (2) | 3.2% (1) | 7.3% (22) |
| **TOTAL COUNTS** | 242 | 30 | 31 | 303 |

Table 10: Distribution of prompt categories and contributors across Train, Val, and Test splits shown as percentages with raw counts in parentheses.

## P  LLM Usage Statement

Large Language Models (LLMs) were used in the experiments for the paper as described, to assist in writing the code to run the experiments, brainstorm the formalization of the task, assist in generating

|  | Unique source texts |
| --- | --- |
| **Train** | 22690 |
| **Validation** | 1391 |
| **Test** | 3809 |

Table 11: Count of unique source texts across Train, Val, and Test datasets.

|  | Human | AI-edited | AI-generated | **Total** |
| --- | --- | --- | --- | --- |
| **Train** | 18810 | 22381 | 18809 | **60000** |
| **Validation** | 753 | 894 | 753 | **2400** |
| **Test** | 1881 | 2238 | 1881 | **6000** |

Table 12: Distribution of examples by label across Train, Val, and Test datasets.

|  |  | Claude Sonnet 4 | GPT-4.1 | Gemini 2.5 Flash |
| --- | --- | --- | --- | --- |
| Train | AI-edited | 7528 | 7521 | 7332 |
| Train | AI-generated | 6205 | 6243 | 6361 |
| Validation | AI-edited | 298 | 298 | 298 |
| Validation | AI-generated | 256 | 235 | 262 |
| Test | AI-edited | 739 | 795 | 704 |
| Test | AI-generated | 627 | 627 | 627 |

Table 13: Distribution of source LLM for AI-edited and AI-generated examples, across Train, Val, and Test datasets.

|  | Train | Validation | Test |
| --- | --- | --- | --- |
| Tone and Style Adjustments | 6477 | 301 | 459 |
| Adding Detail | 4142 | 116 | 426 |
| Fluency and Flow | 2302 | 57 | 301 |
| Structure and Organization | 2110 | 71 | 176 |
| Concision | 2037 | 76 | 301 |
| Paraphrasing | 1679 | 176 | 229 |
| Clarity and Precision | 1229 | 33 | 290 |
| Grammar and Mechanics | 400 | 27 | 70 |
| General Improvement | 2005 | 37 | 0 |

Table 14: Composition of the AI-edited dataset, by split

|  |  | Mean Word Count | Min Word Count | Max Word Count |
| --- | --- | --- | --- | --- |
| **Train** | **Human** | 330.45 | 75 | 799 |
| **Train** | **AI-edited** | 328.11 | 75 | 799 |
| **Train** | **AI-generated** | 302.88 | 76 | 799 |
| **Validation** | **Human** | 331.02 | 75 | 798 |
| **Validation** | **AI-edited** | 324.29 | 75 | 799 |
| **Validation** | **AI-generated** | 299.81 | 79 | 797 |
| **Test** | **Human** | 241.47 | 75 | 799 |
| **Test** | **AI-edited** | 258.37 | 76 | 798 |
| **Test** | **AI-generated** | 218.05 | 76 | 792 |

Table 15: Word count statistics across splits for Human, AI-edited, and AI-generated texts in the dataset.

Table 16: Ternary Classification (Soft N-Grams)

| Model | # buckets | # params | Acc. | Macro F1 | Human F1 | AI F1 | AI-edited F1 |
|---|---|---|---|---|---|---|---|
| Mistral-Small-24B-Base-2501 | 4 | 24B | 0.897 | 0.899 | 0.896 | 0.941 | 0.861 |
| Mistral-Small-24B-Base-2501 | 5 | 24B | 0.873 | 0.876 | 0.913 | 0.875 | 0.840 |
| Mistral-Small-24B-Base-2501 | 6 | 24B | 0.882 | 0.886 | 0.886 | 0.918 | 0.853 |
| Mistral-Small-24B-Base-2501 | Regression | 24B | 0.861 | 0.865 | 0.880 | 0.887 | 0.828 |
| Mistral-Nemo-Base-2407 | 4 | 12B | 0.896 | 0.899 | 0.895 | 0.941 | 0.860 |
| Llama-3.1-8B | 4 | 8B | 0.895 | 0.898 | 0.895 | 0.942 | 0.856 |
| Llama-3.2-3B | 4 | 3B | 0.858 | 0.861 | 0.888 | 0.876 | 0.820 |

the figures for the paper, assist with LaTeX formatting, and review the paper to help the authors with constructive feedback. The authors did **not** use LLMs directly in the original writing of the manuscript, but did use LLMs to help with wording and phrasing in some sections. The authors take full responsibility for the factuality and originality of the content in this manuscript.

## Q  GRAMMARLY SUPERVISION SCORES

In Figures 10 and 11, we show the distributions of scores according to different intermediate supervision metrics by Grammarly edit prompt.

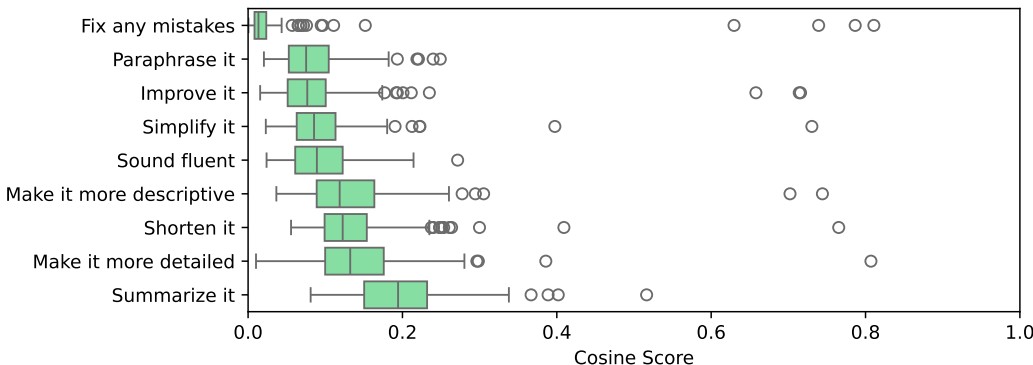

Figure 10: Distribution of cosine scores on dataset from Grammarly by edit instruction.

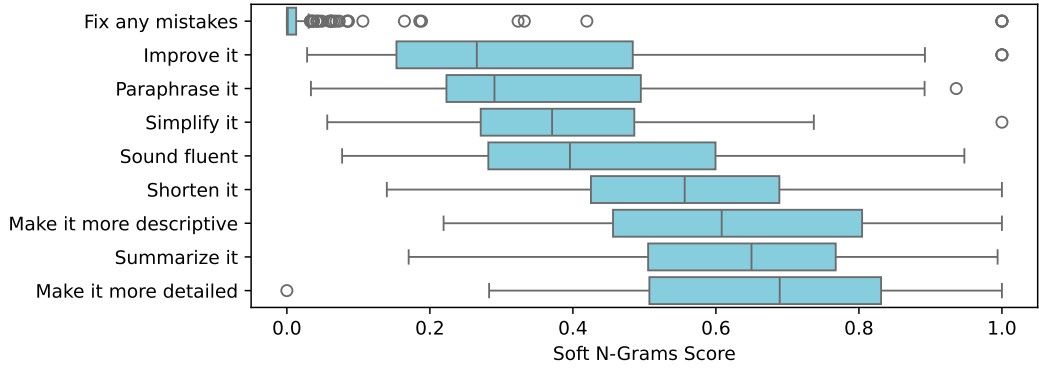

Figure 11: Distribution of soft n-grams scores on dataset from Grammarly by edit instruction.

Table 17: Ternary Classification (Cosine Similarity)

| Model | # buckets | # params | Acc. | Macro F1 | Human F1 | AI F1 | AI-edited F1 |
|---|---|---|---|---|---|---|---|
| Mistral-Small-24B-Base-2501 (selected) | 4 | 24B | 0.902 | 0.904 | 0.904 | 0.941 | 0.868 |
| Mistral-Small-24B-Base-2501 | 5 | 24B | 0.874 | 0.878 | 0.880 | 0.912 | 0.841 |
| Mistral-Small-24B-Base-2501 | 6 | 24B | 0.855 | 0.859 | 0.872 | 0.882 | 0.822 |
| Mistral-Small-24B-Base-2501 | Regression | 24B | 0.857 | 0.861 | 0.883 | 0.879 | 0.820 |
| Mistral-Nemo-Base-2407 | 4 | 12B | 0.889 | 0.896 | 0.900 | 0.939 | 0.845 |
| Llama-3.1-8B | 4 | 8B | 0.861 | 0.865 | 0.877 | 0.899 | 0.819 |
| Llama-3.2-3B | 4 | 3B | 0.852 | 0.855 | 0.883 | 0.878 | 0.804 |

Table 18: Binary Classification (Soft N-Grams)

| Model | # buckets | # params | Human vs Rest Acc. | Human vs Rest F1 | AI vs Rest Acc. | AI vs Rest F1 |
|---|---|---|---|---|---|---|
| Mistral-Small-24B-Base-2501 | 4 | 24B | 93.967 | 95.542 | 94.333 | 90.151 |
| Mistral-Small-24B-Base-2501 | 5 | 24B | 94.400 | 95.857 | 92.983 | 87.533 |
| Mistral-Small-24B-Base-2501 | 6 | 24B | 93.033 | 94.969 | 95.233 | 91.833 |
| Mistral-Small-24B-Base-2501 | Regression | 24B | 92.683 | 94.664 | 93.433 | 88.419 |
| Mistral-Nemo-Base-2407 | 4 | 12B | 93.267 | 94.993 | 96.467 | 94.147 |
| Llama-3.1-8B | 4 | 8B | 93.033 | 94.780 | 96.483 | 94.157 |
| Llama-3.2-3B | 4 | 3B | 92.783 | 94.664 | 93.367 | 88.322 |

Table 19: Binary Classification (Cosine Similarity)

| Model | # buckets | # params | Human vs Rest Acc. | Human vs Rest F1 | AI vs Rest Acc. | AI vs Rest F1 |
|---|---|---|---|---|---|---|
| Mistral-Small-24B-Base-2501 (selected) | 4 | 24B | 93.783 | 95.377 | 96.433 | 94.062 |
| Mistral-Small-24B-Base-2501 | 5 | 24B | 92.583 | 94.608 | 94.867 | 91.170 |
| Mistral-Small-24B-Base-2501 | 6 | 24B | 92.167 | 94.332 | 93.317 | 88.202 |
| Mistral-Small-24B-Base-2501 | Regression | 24B | 92.583 | 94.534 | 93.183 | 87.867 |
| Mistral-Nemo-Base-2407 | 4 | 12B | 93.475 | 94.568 | 96.448 | 93.847 |
| Llama-3.1-8B | 4 | 8B | 92.017 | 94.055 | 94.183 | 89.869 |
| Llama-3.2-3B | 4 | 3B | 92.133 | 94.046 | 93.083 | 87.776 |

