# OpenReview forum: "EditLens: Quantifying the Extent of AI Editing in Text"
_ICLR.cc/2026/Conference — ICLR 2026 Poster_

### Official Review · Reviewer_BnBc · 2025-10-30

**Soundness:** 3
**Presentation:** 3
**Contribution:** 3
**Rating:** 6
**Confidence:** 3

**Summary:**

This paper introduces EditLens, a detector that estimates the degree of AI editing in a text under homogeneous mixed authorship, where human and AI contributions are entangled at the token level—rather than simply classifying text as human- or AI-written. The task is formalized as regressing a change-magnitude score from the edited text alone, without access to the original. Using cosine distance of sentence embeddings and a "soft n-grams" precision metric—validated against human judgments—as supervision, EditLens is trained on a synthetic dataset of human texts and their AI-edited variants generated via diverse prompts and LLMs. The model fine-tunes open LLM backbones with QLoRA using a multi-task objective. EditLens achieves strong performance as a classifier and produces monotonic scores with increasing edit intensity. It generalizes well to unseen LLMs and domains, and a Grammarly case study shows instruction-specific edit distributions. Models and data will be released.

**Strengths:**

- Strong and timely motivation. The paper identifies a real and underexplored gap in AI text detection, i.e., the prevalence of heterogeneous, mixed authorship rather than purely human or AI-generated content. By shifting from binary classification to a continuous estimation of edit intensity, it tackles a far more realistic and societally relevant problem.
- The curated dataset of human texts paired with systematically AI-edited variants offers a rich benchmark for studying varying degrees of human-AI co-authorship. Its structure provides a foundation for future research.
- EditLens demonstrates monotonic response to increasing AI involvement, strong macro-F1 across binary and ternary settings, and clear ablation studies.
- OOD tests and the Grammarly case study provide useful insights into how edit types and prompt categories affect the model's judgments.

**Weaknesses:**

- The dataset is generated exclusively via AI editing human-written text, capturing only one direction of authorship interaction. Yet heterogeneous mixed authorship fundamentally refers to cases where both humans and AI make substantial contributions to the same text. In many real-world workflows, such as researchers revising AI-generated drafts, the reverse direction (human editing AI text) is equally common and important.
- Limited human study scope and unclear methodology. Only three annotators assessed 100 samples using pairwise/tie protocols, without inter-annotator error analysis. Moreover, the paper does not clearly describe the annotation process, such as annotator recruitment or instructions, which limits the reproducibility and interpretability of the results.
- The supervision target for EditLens is derived from embedding and n-gram similarity metrics, which serve only as proxies for the true extent of AI editing. As these same metrics are also used for evaluation, the model may appear to perform well simply because it learns to reproduce the proxy rather than measure real editing strength. A clearer human-grounded calibration or validation study with more annotators would make the construct more reliable and interpretable.
- While the paper is generally clear, some cross-references are vague and hinder readability. For instance, in about line 206, the authors write "see the Appendix", but do not specify which appendix section actually contains it. Such vague pointers make it difficult for readers to locate the intended material, especially given the length of the supplementary text.

**Questions:**

- Beyond pairwise "which is more AI-edited," can you provide an absolute magnitude annotation (e.g., 5- or 10-point scale) with more annotators, report inter-rater reliability, and correlate EditLens with that scale?
- How does EditLens fare under paraphrasers/back-translation/"humanizers" and multi-pass mixed human↔AI edits beyond five steps? Any detection degradation curves?
- Since the supervision labels rely on soft n-gram precision and embedding similarity, how sensitive are the results to different choices of parameters (a,b,τ) or sentence encoders? A brief ablation or robustness analysis would help confirm that the reported improvements are not specific to one proxy configuration.
- Does EditLens generalize to the reverse case where humans edit AI-written drafts? If not, do the authors anticipate differences in the learned signal or dataset construction for such cases?

If the authors can substantively address the questions and weaknesses outlined above, I would be inclined to raise my score.

---

> ### Author Response · Authors · 2025-11-26
>
> Thank you for the positive response and feedback on our work. The reviewer’s main concerns were the proposed method’s potential to generalize to human-edited AI and other transformations, the limited scope of our human evaluation, whether we are using the same metrics to train and eval, and the sensitivity of EditLens to different encoders during the intermediate supervision step. We respond to their comments point-by-point:
>
> ### 1. Directionality of authorship (AI-edited human vs human-edited AI)
>
> While our dataset consists of exclusively fully human, fully AI, or AI-edited human text, we exhibit strong generalization to human-edited AI text. The reviewer might have missed Section 4.7 in our paper, in which we evaluate human-edited AI text, and show via histogram that human editing of AI text decreases scores and shifts the distribution more human (Figures 8 and 9).
>
> ### 2. Additional human study scope
>
> We are currently running an expanded human study with twice as many samples (100 -> 200) and 7 annotations per sample (up from 3). We will have the results before the conclusion of the discussion period and update both the PDF submission and our comment here with the new results.
>
> ### 3. Proxy labels for both training and evaluation
>
>   - We agree this is a source of confusion and will clarify that we explicitly do NOT use the same metrics for training and evaluation.
>   - In Section 4.1 (APT-Eval), we correlate the EditLens score with _prompt_-based editing intensity, e.g., “apply minor polish,” “apply major polish.” We also correlate the EditLens scores with _different_ similarity metrics in the APT-Polish paper: Levenshtein, Jaccard, and semantic similarity metrics are defined differently from our training supervision metric.
>  - In Sections 4.2 and 4.3, any AI editing is its own class. No intermediate metric is needed to perform the relaxed binary and ternary evaluations, so no “training on the test metric” is present.
>  - In Section 4.5, we evaluate the monotonicity of EditLens scores with the number of edits, which is also a metric-independent evaluation.
>  - In Section 4.8, we present a qualitative evaluation on Grammarly, which shows that EditLens scores roughly separate by category/editing prompt. This is also independent of the supervision metric.
>
> ### 4. Robustness/sensitivity to encoder choice - **[NEW EXPERIMENT]**
> Thank you for pointing this out, we have added an additional ablation that uses a different encoder (Qwen3) and will include the results in the revision of our paper:
>
> #### Binary Classification
> - **Best Human F1:** 0.96
> - **Best AI F1:** 0.93
>
> ##### Ternary Classification
> - **Accuracy:** 0.902
> - **Macro F1:** 0.903
>
> **Per-class F1 scores**
> | Class       | F1 Score |
> |-------------|----------|
> | Human       | 0.891    |
> | AI          | 0.959    |
> | AI-Edited   | 0.861    |
>
> We observe no significant difference in results between models trained on cosine distance of different embeddings.
>
> ### 5. Robustness to paraphrasers and humanizers
> The robustness to paraphrasers and humanizers is not a core component of our work because the editing prompts themselves can be thought of as “paraphrasers.”
>
> ### 6. Requested evaluation on multi-pass editing
> We already include a 5-pass AI editing trajectory and show monotonic increases in scores. We agree that multiple passes by AI and humans is an interesting case to study, but do not have the time or resources to collect such a dataset in the review period and thus leave this for future work.
>
> ### 7. Readability and clarification of references
> We commit to clarifying the cross-references to the Appendix and adding stronger pointers for visibility in the revised version of our paper, which will be submitted before the conclusion of the review period.
>
> We hope that these clarifications and commitments to revise our submission are fully satisfactory. If so, we humbly request that you raise your score. Thank you so much for your time and consideration!

---

### Official Review · Reviewer_K4W3 · 2025-10-30

**Soundness:** 4
**Presentation:** 2
**Contribution:** 4
**Rating:** 6
**Confidence:** 4

**Summary:**

The paper introduces a method to predict the extent to which text has been edited by AI. The method is given a piece of text, and must predict (without seein the original unedited source) the extent to which the text was generated by an LLM. This is possible due to the statistical distinctiveness of AI-text compared to human-written text, and is done by training a regression head on top of a transformer, to regress more embedding/n-gram based metrics, which are shown to correlate with expert opinion of AI-editing.
Experimental results seem very strong, though they are presented in such a way that lacks clarity and makes the paper much less valuable in terms of scientific findings, which is unfortunate. But in short, the model can be used in both binary and ternary classification, and is shown to be robust to other kinds of editings that are not included in the training, giving a sense of generalization.

**Strengths:**

- AI detection is an important topic, and the framing is novel and interesting, for example the introduced concepts of heterogeneous and homogeneous edited text, which expand prior simplistic views in the literature about text editing.
- The method and use of data is quite clever and validated through an expert annotation.
- The findings indicate that this approach is very strong, as it not only outperforms at the correlation task, but also at the more boxed "classification" tasks, through the selection of a threshold. It is an elegant finding that "relaxing" the problem helps improving performance on the initial problem.

**Weaknesses:**

Though I am very positive about the work overall, my score is only slightly positive for now. I would be willing to raise my score if the authors commit to improvements in the presentation of the work.
A main limitation of the current submission is the extremely rushed description of results. To name only a few things: (1) APT-Eval is never explained or cited (?) leaving readers unclear of what it is and where it comes from, (2) in 4.3 you mention a "calibration procedure above" that is I believe never defined, (3) Table 2 is never described (?), (4) most results are described in a way that is descriptive but not interpretative: you need to help the reader understand the result and why it is important, not just what it is.
This generally really lowers the value of the submission as is, as a reader can tell that the method is promising (clearly), but is not clear about when it shines and when it doesn't and what leads to the sucess.

I highly recommend picking a subset of the presented findings and going over them adequately, rather than rushing through all of them at breakneck speed.

2. I would also encourage the authors to reflect on the work and describe more scientifically where the method still lags, or where there is still room for improvement. As is, all results describe the EditLens method as "solving" the task to the best extent, which is wonderful, but leaves the reader not knowing what remains unsolved. As the authors of this work, you have this perspective from your deep involvement in this project, and scientific communication requires you to help put the work in perspective. A good paper should not only report promising scientific findings, but also help guide the field forward from there.

**Questions:**

1. The two proxy metrics used for training achieve similar (0.66-0.67) correlation with experts, how well do they correlate with each other. Is there a combination of both (say min(A,B), mean(A,B), max(A,B)) that correlates better with experts? Why do you keep both, I presume they each add something semantic to the problem. Can you comment on why both are needed?
2. It seems like the threshold (based on Table 1) of EditLens on the binary tasks are still rather close to the extremes (0.04 and 0.96), when one could have expected that these would be closer to the mid-range? Can you comment on why that is observed? Does this mean that the community assumed that a 4% change to a piece of human-written text is the threshold at which there should be disclosure of AI-editing?
3. I wonder to what extent the Prompt CLS helps. Does having a model know about categories of edits help it learn to regress, did you conduct any training ablations?
4. Please see questions listed in the weaknesses. How would you modify the current findings /discussion description to improve the submission's narrative?

---

> ### Author Response · Authors · 2025-11-26
>
> Thank you for your positive assessment of our work and the detailed feedback on the presentation. The reviewer’s core concerns are the hurried and confusing organization of our results in the main paper and the lack of contextualization of this project in its field of research. We respond to each of the reviewer’s points below, and commit to updating our submission before the discussion period has ended.
>
> ### 1. Presentation of results
>
>   - We agree that the results are too compressed. We will focus on the main results--**APT-Eval dataset, the binary metrics, the ternary metrics, performance on multi-edited AI text**, and **generalization to human-edited AI text**--in more depth, and move the supplementary results (OOD performance, generalization to AI-edited AI text) to the Appendix.
>   - We apologize for the oversight in defining our calibration procedure and in describing APT-Eval. The procedure was to select thresholds based on a held-out validation set to maximize the F1 score. We then applied those thresholds on the test set. We will add a brief description of this procedure in the revision. We will also add a concise description and citation for APT-Eval.
>   - We will also provide more context around Table 2.
>
> ### 2. Paper narrative and limitations
> Thank you for the suggestion to provide more context around the errors and limitations of the methods and to guide the field forward in future work. With the space gained by moving our supplementary results to the appendix, we will include in-depth discussion of the tradeoffs in our intermediate supervision metrics (see #3 below for details). We will also highlight the potential for improvement of the intermediate supervision metrics as future work.
>
> ### 3. Comparison between cosine and soft n-gram methods
>   - The reason that we study both the cosine and soft n-gram methods is to explore the trade-off between local and global edits. Cosine score reflects the change to the document overall, while soft n-grams measures the number and scope of local edits. We believe that both metrics are complementary and measure different facets of editing, which is why we keep the results for both. We will add ensembling the metrics, and improvement of the intermediate supervision metrics in general, as future work to the discussion section.
>
>   - Cosine score is actually a special case of soft n-grams wherein the span length is set equal to the document length. In order to find a middle ground between the two, the span length in soft n-grams can be varied. We leave this as an implementation detail as we have seen that both extremes in span length (short and document long) produce similar results.
>
> ### 4. Thresholds near extreme values
>   - 0.04 is a scaled similarity score, which is not a literal percentage of changed tokens, so a score of 0.04 should not be interpreted as “4% of the text was changed.”
>   - The reason that the thresholds are near the extremes is because the binary classification tasks are different from the normal “fully AI vs. fully human” AI detection task. We are calibrating the thresholds for the two binary tasks: “human vs. any AI”, which means we count even very lightly AI-edited text as AI in this task. Therefore the threshold for that task should be very low, because lightly edited text has very low EditLens scores. The other task is “human + all AI-edited vs. fully AI-generated,” so even very heavily AI-edited text counts as human in this task, so consequently we should expect the threshold to come out very high.
>
> We hope that these clarifications and commitments to revise our submission are fully satisfactory. If so, we humbly request that you raise your score. Thank you so much for your time and consideration!

---

> ### Author Response · Authors · 2025-11-26
>
> ### 5. Impact of the prompt classification head - **[NEW EXPERIMENT]**
> Thank you for the suggestion to add an ablation on the prompt classification head. We ran this ablation study have included the results here:
>
> ### Cosine Score with no prompt CLS (# with prompt CLS)
> #### Binary Classification
> - **Best Human F1:** 0.96 (0.95)
> - **Best AI F1:** 0.97 (0.94)
>
> #### Ternary Classification
> - **Accuracy:** 0.877 (0.902)
> - **Macro F1:** 0.880 (0.904)
>
> **Per-class F1 scores**
> | Class       | F1 Score |
> |-------------|----------|
> | Human       | 0.833 (0.896)   |
> | AI          | 0.965 (0.941)   |
> | AI-Edited   | 0.841(0.868)   |
>
> ### Soft N-Grams Score with no prompt CLS (# with prompt CLS)
> #### Binary Classification
> - **Best Human F1:** 0.96 (0.96)
> - **Best AI F1:** 0.97 (0.90)
>
> #### Ternary Classification
> - **Accuracy:** 0.901 (0.897)
> - **Macro F1:** 0.904 (0.899)
>
> **Per-class F1 scores**
> | Class       | F1 Score |
> |-------------|----------|
> | Human       | 0.903 (0.896)   |
> | AI          | 0.944 (0.941)   |
> | AI-Edited   | 0.864 (0.861)   |
>
> For the model trained on cosine distance, the binary metrics improved after removal of the prompt classification head, but the ternary metrics suffered. For the model trained on the soft n-grams score, both binary and ternary metrics improved slightly. We suspect that the results are inconclusive because the original formulation of the edit prompt classification task introduced unintended ambiguity: some of the edit prompt categories could be considered overlapping. For simplicity, we will omit the prompt classification task in our revision and propose its development as future work. We are in the process of retraining the models and updating the results in the paper. The revision we will submit before the conclusion of the discussion period will include the updated results in the main paper.
>
> We hope that these clarifications and commitments to revise our submission are fully satisfactory. If so, we humbly request that you raise your score. Thank you so much for your time and consideration!

---

### Official Review · Reviewer_Ri7B · 2025-10-31

**Soundness:** 2
**Presentation:** 3
**Contribution:** 3
**Rating:** 2
**Confidence:** 3

**Summary:**

This paper introduces EditLens, a novel approach for the quantification of the magnitude of AI editing present within a given text. To this aim, the authors first propose a set of similarity metrics to quantify the magnitude of AI editing in a text given the original, human-written one. Second, after validating these metrics via human annotators, the EditLens regression model is trained on such scores to predict the amount of AI editing in unseen texts. Experiments were performed using diverse models and datasets, and a case study on the Grammarly writing assistant is presented to further showcase the EditLens' capabilities.

**Strengths:**

- This paper aims to address an important issue in current AI detection approaches, as simply stating AI vs Human-written is reductive and leads to a lot of false positives in current settings.
- EditLens directly regresses the score corresponding to the degree of AI involvement in a text as a whole, rather than simply returning "human-AI mixed" as in the case of its competing methods.
- Results seems to be promising compared to existing approaches, and the experimental setup covers multiple domain scenarios as well as LLMs.

**Weaknesses:**

- The proposed approach extensively leverages a set of intermediate supervision metrics to train EditLens. However, these represent a key weakness to me. Specifically, these metrics heavily rely on semantic (dis)similarity among the original human-authored texts and their AI-edited counterparts, being potentially misleading. In this regard, if an edit keeps the same semantic meaning of a text while changing multiple "words", the cosine distance would receive a lighter shift than expected. Note that similar considerations hold for the soft n-grams, since they still account for cosine similarity. Related to the previous point, EditLens is hence optimized to follow a concept of similarity learned by *Linq-Embed-Mistral* (which is in charge of producing embeddings), creating a sort of model-dependent similarity, which might include potential biases due to training data of the encoder, as well as capture stylistic variations rather than actual editing.
- The human evaluation process is not sufficiently robust, as the authors relied on only three annotators and evaluated merely 100 tasks. Although the reported agreement is moderate, the small number of annotators and limited sample might impact the generalizability of the reported findings, e.g., due to annotator biases or sample variability.
- EditLens is found to achieve strong correlation with edit magnitude metrics, but isn't it by construction? It is trained on edit metric signals, so this correlation should be expected rather than a finding.

**Questions:**

- Related to the first raised weakness, I wonder if the authors could experiment with similar "attack" scenarios (e.g., similar semantics with tangible edits), to better prove the robustness of the proposed approach. Furthermore, do results change when the encoder is replaced with a similar-performing one? This would better highlight any potential "fitting" of the model biases rather than actual editing.
- I would like to ask the authors how robust EditLens to editing made by humans is. That is, a text that is rephrased by humans, comes with the risk of being flagged as AI edited? Could the authors provide some more insights into false positives?
- Why is EditLens based on soft n-grams found to be better in the binary setting, whereas the cosine backbone performs better in the ternary classification?
- Appendix N has a missing reference to the confusion matrix figure.

---

> ### Author Response · Authors · 2025-11-26
>
> We thank the reviewer for the constructive feedback and for noting the importance of a new paradigm in AI detection that recognizes the full spectrum of human and AI co-authorship. The reviewer’s core concerns were about the sensitivity of the intermediate supervision metrics to semantic-preserving edits, the limitations of the human study, and the robustness of our method to different encoders in the intermediate supervision step. We have run additional experiments to address these concerns and include the results of those new experiments below. We also commit to including these results in the revision of our paper, which we will post to OpenReview in the next couple of days before the discussion period has ended.
>
> ### 1. Intermediate supervision metric relies on semantic dissimilarity - **[NEW EXPERIMENT]**
>
>   - While it is true that Linq-Embed-Mistral embeds semantics, we actually find that the cosine distance for these embeddings does not exclusively encode semantics, and also encodes other semantic-invariant changes, such as stylistic and lexical variations.
>   - With regard to “if an edit keeps the same semantic meaning of a text while changing multiple ‘words’, the cosine distance would receive a lighter shift than expected,” we present additional results where we measure the cosine distance between a source text and a text that has undergone _only synonym replacement_, for the described case where the overall semantics stay the same and only the words have been changed. Among 13,000 texts from the RAID benchmark (Dugan et. al. 2024) which have undergone only synonym replacement, we find the following distribution of bucket labels for the task:
> | Bucket Label          | Percentage | Count (out of 13,371) |
> |-----------------------|------------|------------------------|
> | Very light / No edit  | 18.93%     | 2,531                  |
> | Light edit            | 46.43%     | 6,208                  |
> | Medium edit           | 24.48%     | 3,273                  |
> | Heavy edit            | 10.16%     | 1,359                  |
>
>   - We can conclude that synonym replacement alone is enough to correspond to significant editing in a nontrivial number of cases. Therefore, it is incorrect to say that Linq-Embed-Mistral “heavily relies on semantic dissimilarity.”
>
> ### 2. Results are dependent on particular embedding choice, Linq-Embed-Mistral - **[NEW EXPERIMENT]**
>   - This is a great point and we ran an additional ablation study with Qwen3 embeddings as supervision. We will add these results to the appendix:
>
> #### Binary Classification
> - **Best Human F1:** 0.96
> - **Best AI F1:** 0.93
>
> ##### Ternary Classification
> - **Accuracy:** 0.902
> - **Macro F1:** 0.903
>
> **Per-class F1 scores**
> | Class       | F1 Score |
> |-------------|----------|
> | Human       | 0.891    |
> | AI          | 0.959    |
> | AI-Edited   | 0.861    |
>
> We observe no significant difference in results between models trained on cosine distance of different embeddings.
>
> ### 3. Insufficient human study - **[NEW EXPERIMENT]**
>
> We are currently running an expanded human study with twice as many samples (100 -> 200) and 7 annotations per sample (up from 3). We will have the results before the conclusion of the discussion period and update both the PDF submission and our comments here with the new results.
>
> ### 4. Performance on human-edited human text - **[NEW EXPERIMENT]**
>
> We conducted an additional evaluation of EditLens on human-edited human text from the ArgRewrite V.2 dataset by Kashefi et al. (2022). We find that the average EditLens score (on a 0-1 scale) on human-edited human text is 0.012. The average EditLens score of human text 0.009, and the average EditLens score of AI-edited human text on the same data is 0.86.
> We conclude from these results that EditLens does not predict human-edited human text as AI-edited. **Human-edited human text is predicted effectively the same as human text by EditLens.**
>
> ### 5. Correlation with edit magnitude is “by construction” and not a finding
>
> For our regression evaluation, some degree of correlation is indeed expected. However:
>
>   - The model never sees the source x at test time; it only sees y. Achieving high correlation with ∆(x,y) from y alone is not trivial and is precisely the modeling challenge we tackle.
>
>   - Our main _detection_ results (binary/ternary) are **not** evaluated against the similarity metrics, but against ground-truth human vs AI vs AI-edited labels induced by the data generation process (Tables 1 and 2). The reported gains (e.g., macro-F1 improvements over Pangram and GPTZero) cannot be attributed to “by construction” effects of the proxy.
>
> - On APT-Eval we correlate with **external similarity metrics** (semantic similarity, Levenshtein, Jaccard) that are not used during training, further mitigating “self-evaluation” concerns.
>
> We will clarify this distinction between _supervision source_ and _evaluation metrics_ in the text.

---

> ### Author Response · Authors · 2025-11-26
>
> ### 6. Soft N-Grams vs. Cosine Score discrepancy
>
> The main difference between Soft N-grams and Cosine Score is that Soft N-Grams is more sensitive to local edits and Cosine Score is more similar to global edits. We hypothesize that Soft N-Grams is better at the binary tasks because it is more sensitive to picking up any change, which is important at distinguishing “human vs. any AI” even if the AI editing is very light. However, Cosine Score is better for distinguishing global edits, which is why it is better at distinguishing mixed text from both “pure” human and AI categories.
>
> We hope that these clarifications and commitments to revise our submission are fully satisfactory. If so, we humbly request that you raise your score. Thank you so much for your time and consideration!

---

### Official Review · Reviewer_yUTu · 2025-11-01

**Soundness:** 2
**Presentation:** 3
**Contribution:** 2
**Rating:** 4
**Confidence:** 3

**Summary:**

This paper addresses the detection of AI-edited text, moving beyond simple classification to quantify the extent of AI involvement. The authors introduce EDITLENS, a regression model trained to predict a continuous score representing the magnitude of AI editing. To achieve this, they create a large-scale dataset of human texts edited by various LLMs and use lightweight similarity metrics, validated against human judgments, as supervision. Their model achieves state-of-the-art performance when adapted for both binary and ternary classification tasks and demonstrates a nuanced ability to track the intensity of AI polish.

**Strengths:**

+ The paper addresses a highly relevant and timely problem. The shift from binary AI detection to quantifying the continuous spectrum of AI co-writing is a necessary and practical step forward for the field, with clear applications in areas like academic integrity.

+ The creation and planned release of a large-scale dataset specifically for the task of detecting homogeneous mixed authorship is a significant contribution. This resource will undoubtedly enable further research in this under-explored area.

+ The paper is exceptionally well-written, logically structured, and easy to follow. The figures are effective at illustrating the core concepts and the model's performance.

**Weaknesses:**

- The paper's fundamental limitation is that EDITLENS is not directly trained to detect the "extent of AI editing," but rather to regress a similarity-driven target (e.g., cosine distance). The authors themselves report only "moderate agreement" (Krippendorff's α ≈ 0.67) between this proxy metric and human judgments. The model is learning to approximate a text similarity function, which may not capture the specific stylistic artifacts and semantic patterns characteristic of AI editing, but rather any significant textual change.

- The study fails to include a crucial control experiment to disentangle the detection of "AI editing" from the detection of "significant editing" in general. The model is exclusively trained on human text edited by AI. A necessary test would be to evaluate its scores on human-written texts that have been substantially edited by humans.

- While the problem formulation is a valuable contribution, the technical novelty of the proposed method is low. The approach consists of fine-tuning a standard language model with a regression head, a well-established technique. The success of EDITLENS is therefore entirely dependent on the quality of the supervision signal, which, as noted in the first point, is a flawed proxy for the actual phenomenon of interest.

**Questions:**

- Given the moderate agreement between your similarity metrics and human annotators, how can you be confident that the model is learning to detect stylistic cues of AI editing, rather than simply learning to approximate a text similarity function? What specific artifacts of AI editing do you believe the model is capturing that a generic similarity metric would not?

- How would you expect EDITLENS to perform on a text written by a human and then heavily edited by another human (e.g., a student paper revised by a writing tutor)? Would the model assign a high AI-edit score in this case? Answering this seems crucial to understanding if the model is a true "AI editor" detector or a more general "heavy revision" detector.

---

> ### Author Response · Authors · 2025-11-26
>
> We thank the reviewer for the acknowledgment of the relevance and timeliness of our work, as well as for their insightful questions. The core concerns are that EditLens is not an “AI editing” detector but a “general change” detector, a needed control for human-edited human text, and low technical novelty. We respond to these points below and will further clarify in the revision of the paper, which we plan to upload in a few days before the conclusion of the discussion period.
>
> ### 1. Is EditLens an “AI edit” detector or a general “any change” detector?
>
> We agree that the model is supervised on similarity-driven targets and we do not claim to directly observe “true” AI authorship. Our goal is to approximate _edit magnitude under a similarity functional that is calibrated to human perception of AI editing_.
>
>   - **Single-input prediction**: The supervision is computed from (x, y), but at inference the model only sees y. Thus, even if the target is derived from similarity, the learning problem is nontrivial: EditLens must infer how far y likely is from a plausible human source purely from surface and stylistic cues. This is fundamentally different from directly evaluating sim(x, y) at test time. Furthermore, only AI edits have positive supervision. Human-edited human text is implicitly present in the training set and supervised with a label of 0, isolating the learning problem to only detecting AI edits.
>
>    - **Evidence that the model is not simply a proxy for “big distance”**:
>
>       * On **human-edited AI text (BEEMO)**, heavier human edits increase the distance between model output and human revision, yet EDITLENS scores decrease on 88.9% of documents and are negatively correlated with both cosine and soft n-gram distances (detailed in §4.7 and Appendix J). This shows that the model does not treat “more editing” as “more AI”; instead, it tends to assign lower AI-edit scores to human revisions, even when they are large in magnitude.
>
>       * On **AI-edited AI text** (§4.6), a single AI edit to an originally human text increases the score by +0.38 on average, whereas an AI edit to an already AI-generated text changes the score by −0.05 on average. Again, this indicates that the model is not simply a function of edit distance but is sensitive to the direction and type of editing.
>
>       * Qualitatively, we observe that high scores are associated with characteristic AI patterns, e.g., over-regularization of style, removal of idiosyncratic phrasing, and insertion of generic connective templates, rather than just large rewrites.}
>
> ### 2. Human edited human text control
>
> Thank you for the constructive suggestion to include a human-edited human text control. We added an additional section to evaluate EditLens on human-edited human text from the ArgRewrite V.2 dataset by Kashefi et al. (2022).
>
>   * We find that the average EditLens score (on a 0-1 scale) on human-edited human text is 0.012. The average EditLens score of human text 0.009, and the average EditLens score of AI-edited human text on the same dataset is 0.86.
>
>  * We conclude from these results that EditLens does not predict human-edited human text as AI-edited. **Human-edited human text is predicted effectively the same as human text by EditLens.**
>
> ### 3. Technical novelty
>
> We agree the regression head itself is standard. The contributions we see as novel are:
>
>   - **Problem formulation**: We rigorously define homogeneous mixed authorship and propose a single-input regression task over edit magnitude, which differs from token-level boundary detection and ternary classification in assumptions, outputs, and supervision (§2, App. A).
>
>   - **Dataset and supervision design**: Creating a large-scale, taxonomy-structured AI-edit dataset and validating lightweight similarity functionals against expert annotators is, to our knowledge, new for this domain.
>
>   - **Empirical finding**: A model trained purely on similarity-derived targets can (i) outperform dedicated binary/ternary detectors, and (ii) generalize in nuanced ways (OOD domains, AI-edited AI text, human-edited AI text). We will make these conceptual points more explicit in the introduction and discussion sections.
>
> We hope that these clarifications and commitments to revise our submission are fully satisfactory. If so, we humbly request that you raise your score. Thank you so much for your time and consideration!

---

### Author Response · Authors · 2025-12-03
**Author response to reviewer concerns & questions**

We thank the reviewers for their acknowledgement of the impact of this work and for the constructive feedback. We would like to summarize and address the most common criticisms pointed out by the reviewers here.
### 1. Human-Edited Human Control - **[NEW EXPERIMENT]**
  - Two of the reviewers pointed out that the study is lacking a control experiment on human-edited human text that shows that EditLens specifically detects AI edits and not edits of any kind. In response, we added an additional section (4.4) to evaluate EditLens on human-edited human text from the ArgRewrite V.2 dataset (Kashefi et al., 2022).
    - We find that the average EditLens score (on a 0-1 scale) on human-edited human text is 0.012. The average EditLens score of human text 0.009, and the average EditLens score of AI-edited human text on the same dataset is 0.86.
  - We conclude that EditLens does not predict human-edited human text as AI-edited. **Human-edited human text is predicted effectively the same as human text by EditLens.**
  - This result is because human-edited human text is implicitly present in the human-labeled portion of our dataset (e.g. news articles go through rounds of human editing before publication).
### 2. Extended human study **[EXPERIMENT IN PROGRESS]**
  - The reviewers pointed out that our human study was limited in scope (only 3 annotators and 100 texts), and that agreement between annotators and our supervision metric was only moderate.
  - We are running an extended human study with 7 annotators and 200 texts in order to get a larger sample size and will update the paper when complete.
  - Disagreements between the annotators and the metric are not necessarily due to the fact that the metric is “incorrect.” The task is inherently subjective. Also, humans are biased by the fact that some edits may be small in scope but easy to perceive. For example, the addition of em-dashes is highly visible and characteristic of AI but it is a small edit in the sense that it does not change the meaning of the text and is mostly cosmetic.
### 3. Clarification that Learning Task is Not Equivalent to Learning a Similarity Function
  - Reviewers yUTu and Ri7b claim the model is only “learning to approximate a similarity function” (yUTu) and that the results correlating external metrics to the EditLens score is trivial/expected because it is “by construction” (Ri7b).
  - We would like to correct this misunderstanding. The similarity functions (cosine score and soft N-grams) are only used at training time to generate the supervision. At inference time, the model does not have access to the source document: EditLens must infer the extent of AI-editing from the edited document alone using surface, stylistic and semantic features: NOT comparing it with the source, which is unknown at test time and not provided to the model as an input.
### 4. Embedding Model Ablation - **[NEW EXPERIMENT]**
  - Reviewers Ri7b and K4W3 asked how sensitive EditLens is to the particular choice of embedding model and hyperparameters used to compute the cosine similarity.
  - We added an ablation swapping out the Mistral-Linq embedding model with Qwen3 embeddings and found negligible differences in the end-to-end evaluation on the metric-independent binary and ternary evaluation tasks (see comment to Reviewer Ri7B for results)
### 5. Supervision Metrics Only Encode Semantic Similarity and not Other Kinds of Edits - **[NEW EXPERIMENT]**
  - Reviewer Ri7b states the key weakness is that the supervision relies on semantic dissimilarity between source and target: i.e., “an edit keeps the same semantic meaning of a text while changing multiple "words", the cosine distance would receive a lighter shift than expected”.
  - We believe that this is a misunderstanding of the information encoded by the embedding. The embedding encapsulates all forms of editing, not just semantic changes.
  - To confirm this, we present additional results where we measure the cosine distance between a source text and a text that has undergone only synonym replacement, for the described case where the overall semantics stay the same and only the words have been changed.
  - Among 13,000 texts from the RAID benchmark (Dugan et. al. 2024) which have undergone only synonym replacement, we find the following distribution of bucket labels for the task:
| Bucket Label          | Percentage | Count (out of 13,371) |
|-----------------------|------------|------------------------|
| Very light / No edit  | 18.93%     | 2,531                  |
| Light edit            | 46.43%     | 6,208                  |
| Medium edit           | 24.48%     | 3,273                  |
| Heavy edit            | 10.16%     | 1,359                  |
  - We can conclude that synonym replacement alone is enough to correspond to significant editing in a nontrivial number of cases. Therefore, it is incorrect to say that Linq-Embed-Mistral “heavily relies on semantic dissimilarity.”

---

### Meta-Review · Area_Chair_TfTR · 2026-01-07

**Summary:**

- No comparison against human-edited human-written text or human-edited AI-written text (yUTu, Ri7B, BnBc)
- Lack of technical novelty (yUTu)
- Intermediate supervision as both the training signal and the evaluation is a bad idea; supervisionmight be fixed on the Mistral embedding model (Ri7B, BnBc)
- Limited human evaluation (Ri7B, BnBc)
- Poor writing with respect to the results (K4W3)

**Reviewer Concerns:**

- Control against human-edited text was well addressed.
- Tried a different embedding model
- Doubled the samples and annotators

I think the authors meaningfully address all concerns.

**Reviewer Scores:**

I think all reviewers would improve their scores.

---

### Decision · Program_Chairs · 2026-01-26

Accept (Poster)